

# Rainbow chains and numerical renormalisation group for accurate chiral conformal spectra

**Attila Szabó**

Department of Physics, University of Zürich, Zürich, Switzerland

attila.szabo@physik.uzh.ch

## Abstract

Based on the relationship between reduced and thermal density matrices in conformal field theory (CFT), we show that the entanglement spectrum of a conformal critical chain with exponentially decaying terms consists of conformal towers of the associated *chiral* CFT, with only weak finite-size effects. Through free-fermion and interacting examples, we show that these entanglement spectra present a reliable method to extract detailed CFT spectra from single wave functions without access to the parent Hamiltonian. We complement our method with a Wilsonian numerical renormalisation group algorithm for solving interacting, exponentially decaying chain Hamiltonians.


---

# 1  Introduction

Over the past decades, studying entanglement properties of quantum many-body states has generated many new insights in condensed-matter theory. In this quest, the notion of the *entanglement spectrum* [1] has proven particularly useful: It is obtained by interpreting the reduced density matrix of a subsystem $A$ as the thermal density matrix of an *entanglement Hamiltonian* $H_{\text{ent},A}$, defined through

$$\rho_A \equiv \exp(-2\pi H_{\text{ent},A}) = \sum_\alpha e^{-\lambda_\alpha}|\alpha_A\rangle\langle\alpha_A|, \qquad H_{\text{ent},A} = \sum_\alpha \frac{\lambda_\alpha}{2\pi}|\alpha_A\rangle\langle\alpha_A|, \qquad (1)$$

where $e^{-\lambda_\alpha}$ and $|\alpha_A\rangle$ are the Schmidt values and vectors; the entanglement spectrum is conventionally defined as the set $\{\lambda_\alpha\}$. The "low-energy" structure of the entanglement spectrum (that is, the structure of leading Schmidt values) contains fingerprints of exotic physics, which would be challenging to diagnose with conventional tools. For example, the entanglement spectrum of a symmetry-protected topological (SPT) state contains degeneracies that reflect the symmetry group that protects the phase [2,3], while the entanglement spectrum of two-dimensional fractional quantum Hall [1] and other chiral topologically ordered [4] states reflect the spectrum of the chiral conformal field theories (CFTs) that describe their edge spectra. The latter, combined with momentum-resolved entanglement spectra from MPS [5] or PEPS [6] simulations on finite cylinders, has become the method of choice to pinpoint chiral spin-liquid phases in numerics [7–11].

Remarkably, entanglement spectra of conformally critical one-dimensional chains show the same structure: While these are necessarily achiral [12], their entanglement spectrum matches that of a *boundary* CFT (BCFT) [13–15], whose spectra, like those of chiral CFTs, are generated by a single set of Virasoro generators [16]. This allows us to obtain chiral conformal spectra (albeit without a clear sense of chirality) using purely one-dimensional simulations, as well as to study the CFTs underlying one-dimensional critical states, even without access to the parent Hamiltonian.

Extracting detailed information about the underlying chiral CFT is, however, challenging for both of these approaches. In two dimensions, the area law of entanglement entropy [17] limits DMRG simulations to narrow cylinders, on which CFT multiplets quickly broaden and dissolve in a continuum of high-entanglement-energy states. On critical chains of length $L$, in line with the logarithmic scaling of entanglement entropy [18], finite-size deviations from the expected spectrum scale as powers of $\log L$, requiring simulations on excessively long chains to suppress them.

In this paper, we propose an alternative approach, effectively reversing the relationship between entanglement spectra and BCFTs. Namely, we construct one-dimensional Hamiltonians whose entanglement spectra match those of a BCFT on a segment of length $\sim L$ with uniformly distributed sites, thereby much improving their finite-size scaling. We find that Hamiltonian terms in the bulk of such a chain decay exponentially from its centre, similar to the so-called *rainbow chains* [19–22], which have been proposed as counterexamples to the entanglement area law in one dimension. While the volume-law scaling of entanglement entropy should generally be concerning for MPS simulations, we find that high-quality MPS representations can be built even for long free-fermion chains, allowing us to recover entanglement spectra in much more detail than it would be possible in computationally viable uniform chains.

For interacting theories, we also design a numerical renormalisation-group (NRG) scheme, based on Wilson's NRG for the Kondo problem [23, 24], to obtain the ground state and low-lying excited states of these rainbow chains, a highly challenging task for variational optimisation methods like DMRG. We obtain these eigenstates in a nonstandard MPS form tailored to their entanglement structure, obviating potential difficulties due to volume-law entanglement. We also show how to convert this representation to a standard MPS, thereby obtaining their entanglement spectra, which again recover CFT predictions to excellent accuracy.

We believe that these approaches are applicable to any one-dimensional conformally critical Hamiltonian, as well as CFT wave functions without known parent Hamiltonians. Furthermore, the structure of the NRG flow and fixed points is likely to contain CFT data beyond BCFT spectra, which would allow us to fully characterise CFTs, even where wave functions are obtained from, for example, parton constructions, rather than a known parent Hamiltonian.

In Sec. 2, we review the connection between CFT entanglement spectra and BCFTs [13,14] and explain our construction of rainbow chains. We demonstrate the power of the approach to obtain accurate spectra for free-fermion CFTs in Sec. 3. We introduce the NRG algorithm in Sec. 4 and demonstrate it for the three-state Potts model in Sec. 4.1. We finish by discussing our results and potential further applications in Sec. 5.

## 2 Conformal transformations and rainbow chains

While the entanglement Hamiltonian (1) does not necessarily match the Hamiltonian of the system, they are closely related at one-dimensional conformal critical points [14]. The reduced density matrix of a quantum field theory is represented by a path integral with a branch cut across the subsystem in question: For 2D CFTs, the flexibility of 2D conformal transformations often allows this geometry to be mapped onto a rectangle, that is, the thermal density matrix of a finite segment of the CFT. That is, the entanglement Hamiltonian is equivalent to the Hamiltonian of the CFT itself, restricted to a segment with some boundary conditions. In contrast to CFTs on closed manifolds, the left- and right-moving modes of such a *boundary CFT* (BCFT) are coupled by the boundary conditions, so their spectrum is generated by only one set of Virasoro generators, similar to the corresponding chiral CFT [16].

In the particular case of the ground state of the finite open segment $\mathrm{Re}\, z \in (-\pi/4, \pi/4)$ with an entanglement cut at the origin [Fig. 1(a)], such a mapping is given by [14]

$$w = \log \tan z \,. \tag{2}$$

The mapping must be regularised by excluding a small circle $|z| < r$ from the domain of the path integral; discretising the theory to a chain of $L$ sites automatically achieves this, with a radius $r \sim 1/L$. It follows that the entanglement spectrum matches that of the BCFT on a segment of length $w_0 \simeq -\log r \simeq \log L$, at inverse temperature $\beta = 2\pi$: this leads to finite-size effects that scale with $\log L$, as observed in Refs. [13,15]. Obtaining detailed and accurate information about the underlying CFT with this method would, therefore, require impractically long chains.

**Inhomogeneous spin chains.** Conformal critical points, however, also arise in spin chains with spatially inhomogeneous Hamiltonian terms. Consider the Hamiltonian

$$H = \sum_{i=1}^{L} f(i) h_i^{(1)} + \sum_{i=1}^{L-1} f(i+1/2) h_{i,i+1}^{(2)} \,, \tag{3}$$

where $f$ is a sufficiently slowly varying function and the one- and two-body Hamiltonian terms $h^{(1)}, h^{(2)}$ are chosen such that the uniform chain $f \equiv 1$ is at a conformal critical point. The

continuum limit of such an inhomogeneous chain is

$$H_{\text{cont}} \simeq \int dx\, f(x) h_{\text{CFT}}(x),$$

(4)

where $h_{\text{CFT}}(x)$ is the Hamiltonian density of the CFT realised by the uniform chain $f \equiv 1$, with speed of light $v_0$. Locally, the Hamiltonian (4) describes the same CFT, albeit with a space-dependent speed of light $v(x) = f(x)v_0$ [22, 25]. This complication can be eliminated by redefining the space coordinate as [25]

$$\tilde{x}(x) = \int^x \frac{dx'}{f(x')} \iff f(x) = \left(\frac{d\tilde{x}}{dx}\right)^{-1}.$$

(5)

Since $\tilde{x}$ measures the time it takes for an excitation travelling at speed $v(x')$ to reach position $x$, the speed of light of the continuum theory is uniformly $v_0$ in $(\tilde{x}, \tau)$-space.[1] Eq. (3) can thus be viewed as a lattice discretisation of the uniform CFT $h_{\text{CFT}}$ using unevenly distributed lattice sites, located at $\tilde{x}(i)$.

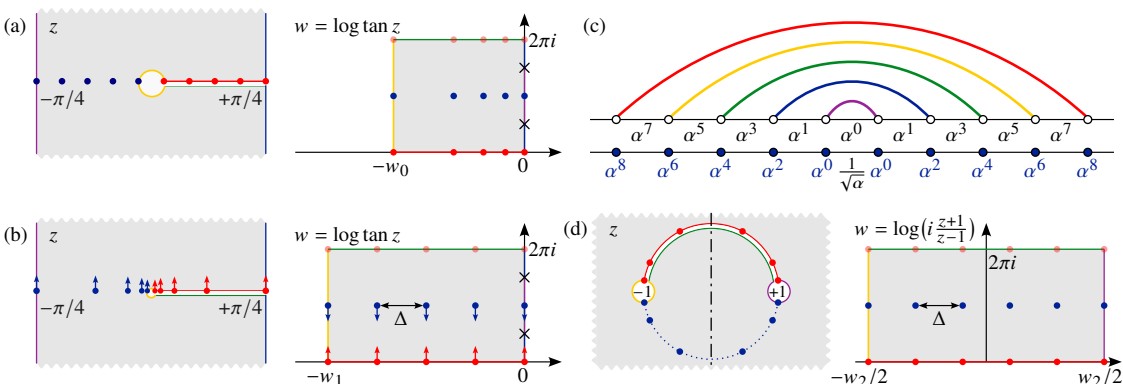

Figure 1: **(a)** The reduced density matrix of an $L$-site chain at a conformal critical point is given by the partition function of the corresponding BCFT on the grey manifold (left). This can be mapped on a BCFT thermal density matrix with system size $w_0 \sim \log L$ using the conformal transformation (2). Therefore, finite-size effects scale with $\log L$, requiring immense system sizes to eliminate them. **(b)** If the sites are instead distributed at uniform spacings $\Delta$ in the *transformed* space, the entanglement spectrum corresponds to a thermal density matrix at system size $w_1 \sim L$. In the original geometry, this corresponds to non-uniformly spaced lattice sites, which can be emulated using position-dependent Hamiltonian terms. **(c)** The rainbow chain is a free-fermion tight-binding chain with exponentially decaying couplings (top line): in the limit $\alpha \to 0$, the ground state is a singlet on each arc of the rainbow, resulting in volume-law entanglement. In models with on-site couplings, the latter also decay exponentially (bottom line). In the limit $w \ll 0$, the conformally deformed chain (b) is identical to a rainbow chain with $\alpha = e^{-\Delta/2}$. **(d)** CFT reduced density matrices in periodic boundary conditions can also be mapped onto thermal density matrices, using the conformal transformation (10). (Time runs radially in $z$-space.) The preimage of sites distributed uniformly in $w$-space between $\pm w_2/2$ remains symmetric under reflection across the imaginary axis. Reflection eigenvalues of Schmidt states can be used to distinguish descendant states with even and odd momentum.

---

[1]We also use units in which $v_0 \equiv 1$ to make the spacetime conformal symmetry explicit.

**Constructing the rainbow chain.** We can thus discretise a continuum CFT into a critical spin chain using an arbitrary set of reference points $\tilde{x}(i)$, at the cost of rescaling the Hamiltonian terms in the spin chain (3). To improve the finite-size scaling of the entanglement spectrum in particular, we would like to use reference points that are uniformly distributed *after* applying the conformal transformation (2) [Fig. 1(b)]. If the post-mapping sites are located at $w_n = -n\Delta$ ($n = 0, 1, \ldots, L/2 - 1$), their preimages are at

$$\tilde{x}(L-n) = -\tilde{x}(n+1) = \arctan e^{-n\Delta} \qquad (0 \le n \le L/2 - 1). \tag{6}$$

By (5), such a layout of sites corresponds to rescaling the Hamiltonian terms by a factor of

$$f(L-n) = f(n+1) = \left(\frac{\mathrm{d}\tilde{x}(L-n)}{\mathrm{d}n}\right)^{-1} \propto \left(\frac{\mathrm{d}z}{\mathrm{d}w}\right)^{-1}_{w=w_n} = 2\cosh(n\Delta) \qquad (0 \le n \le L/2-1). \tag{7}$$

Since $n$ is largest in the middle of the chain, the largest Hamiltonian terms appear there. For sufficiently large $L\Delta$, $2\cosh(n\Delta) \approx e^{n\Delta}$ in the middle section of the chain, i.e., the Hamiltonian terms decay exponentially going away from the centre of the chain with decay rate $\Delta$.

We could in fact consider chains with Hamiltonian terms that decay exponentially throughout [Fig. 1(c)]:

$$f(L/2 - x) = f(L/2 + 1 + x) = e^{-\Delta x} = \alpha^{2x} \qquad (x \ge 0, \alpha := e^{-\Delta/2}). \tag{8}$$

Such chains, dubbed *rainbow chains* on account of the structure of their ground states [Fig. 1(c)], have already been proposed in the literature [19–22] as a counterexample to the area law of entanglement in local Hamiltonians [17, 26]. As we show later, the behaviour of the Hamiltonian does not change significantly between the deformations (7) and (8). In the limit $L \to \infty$, the rainbow chain would in fact correspond to discretising the CFT Hamiltonian over the infinite line using $\tilde{x}(\pm n) = \pm e^{n\Delta}$ as reference points. This makes rainbow chains ideally suited for the numerical renormalisation group procedure in Sec. 4, whose steps can be viewed as dilatation by a factor of $e^\Delta$, resulting in a physically motivated fixed point.

Finally, we need to fix the scale factor $f$ for the central bond, which has not been addressed so far. For chains with only two-body terms, we found empirically that evaluating (7, 8) at the position of the centremost *sites* leads to the best convergence. That is, we set

$$f\left(\frac{L+1}{2}\right) = \begin{cases} 2\cosh[(L/2-1)\Delta], & \text{conformal chain (7),} \\ 1, & \text{rainbow chain (8),} \end{cases} \tag{9a}$$

the latter of which matches the prescription of Refs. [19–22] for free-fermion rainbow chains. For Hamiltonians with both one- and two-site terms, we find the correct prescription by considering the transverse-field Ising model, which can be mapped on a tight-binding model of a doubled number of Majorana fermions (Sec. 3.2). To match the Hamiltonian term scaling (7), the effective positions of these Majorana sites become $\tilde{x}(n \pm 1/4)$. Evaluating (7, 8) at the centremost of these sites, $L + 1/4$ or $L + 3/4$, we get

$$f\left(\frac{L+1}{2}\right) = \begin{cases} 2\cosh[(L/2-3/4)\Delta], & \text{conformal chain (7),} \\ \alpha^{-1/2}, & \text{rainbow chain (8).} \end{cases} \tag{9b}$$

**Periodic boundary conditions: Reflection symmetry as a proxy for momentum.** In addition to open chains, we may also consider conformally critical systems in periodic boundary conditions. These are most naturally represented as a CFT path integral over the whole plane: Imaginary time runs radially, so the reduced density matrix of one half of the ring is obtained

using a branch cut along a half-circle [Fig. 1(d)]. This geometry can also be mapped onto a thermal density matrix using the conformal map

$$w = \log\left(i\frac{z+1}{z-1}\right); \tag{10a}$$

on the unit circle $z = e^{i\theta}$ (that is, at imaginary time $\tau = 0$), this reduces to

$$w = -\log\tan\frac{\theta}{2}. \tag{10b}$$

Similar to the case of open boundary conditions, we consider rainbow-deformed rings where the lattice sites are mapped by (10) to $w_i$ at uniform spacing $\Delta$. For a chain of even length $L$, these sites are at

$$w_n = \left(n - \frac{L+2}{4}\right)\Delta = -\frac{L}{4} + \frac{1}{2}, \dots, \frac{L}{4} - \frac{1}{2},$$
$$\theta(n) = \theta(L/2 + n) - \pi = 2\arctan e^{w_n} \qquad (1 \le n \le L/2). \tag{11}$$

From (5), we then obtain the following prefactors to the Hamiltonian terms:

$$f(n) = f(L/2 + n) = \left(\frac{d\theta}{dw}\right)^{-1} = \cosh w_n \qquad (1 \le n \le L/2),$$
$$f(1/2) = f\left(\frac{L+1}{2}\right) = \begin{cases} \cosh w_1 = \cosh\left(\frac{L-2}{4}\Delta\right), & \text{two-site terms only,} \\ \cosh\left(\frac{L-1}{4}\Delta\right), & \text{one- and two-site terms;} \end{cases} \tag{12}$$

the Hamiltonian terms across the two entanglement cuts are handled analogously to (9).

The key difference between open and periodic boundary conditions is that the post-mapping lattice sites $w_i$ are laid out symmetrically at positive and negative values of $w$. Even before the mapping, this implies an *exact* mirror symmetry of the Hamiltonian across the imaginary axis, which also requires that all Schmidt vectors be mirror symmetric. Importantly, the symmetry eigenvalue $\pm 1$ can be used to distinguish states that would have even and odd angular momentum in the corresponding chiral CFT: In a BCFT with free boundary conditions, the Virasoro generators $L_{-n}$ are themselves mirror symmetric, with eigenvalue $+1$ $(-1)$ for even (odd) $n$; the mirror eigenvalue of a descendant state thus depends on the total parity of Virasoro generators needed to reach it, that is, the parity of its angular momentum.

## 3 Free-fermion examples

We first illustrate the ideas outlined above through the ground-state entanglement spectra of quadratic fermionic Hamiltonians. These can be computed using only the spectrum of the single-particle Green's function [27], which can be obtained at modest computational cost even for long chains with coupling strengths spanning several orders of magnitude. Furthermore, as every Schmidt vector of a Slater-determinant (Pfaffian) state is itself a Slater determinant (Pfaffian) [28–31], accurate MPS representations of these wave functions can also be built without the need for computationally expensive variational optimisation (e.g., using DMRG) [31–34].

In particular, we consider the nearest-neighbour tight-binding model and the Kitaev chain at the topological transition, the latter of which is equivalent to a nearest-neighbour quadratic Hamiltonian of Majorana fermions. Upon the Jordan–Wigner transformation

$$\sigma_i^+ = (-1)^{\sum^{i-1} n_j} c_i^\dagger, \qquad \sigma_i^- = (-1)^{\sum^{i-1} n_j} c_i, \qquad \sigma^z = 2c_i^\dagger c_i - 1, \tag{13}$$

these are also equivalent to a nearest-neighbour XY model and the critical transverse-field Ising (TFI) model, respectively. For both models, we consider three geometries and choices of the inhomogeneity $f(x)$:

- We refer to the open chain with $f(x)$ given by (7) as the *conformal chain*, as its entanglement spectrum is expected to recover the energy spectrum of a uniform chain under the conformal transformation (2) [Fig. 1(b), see also Appendix A].

- Likewise, we call the periodic ring with $f(x)$ given by (12) the *conformal ring* [Fig. 1(d)].

- Finally, we consider open chains with exponentially decaying $f(x)$ (8), which we call *rainbow chains*.

## 3.1 Complex fermions (XY chain)

We first consider the nearest-neighbour tight-binding chain

$$H = -J \sum_i f(i + 1/2) \left( c_i^\dagger c_{i+1} + c_{i+1}^\dagger c_i \right), \tag{14a}$$

or, equivalently, the nearest-neighbour ferromagnetic XY chain

$$H = -J \sum_i f(i + 1/2) \left( \sigma_i^+ \sigma_{i+1}^- + \sigma_i^- \sigma_{i+1}^+ \right). \tag{14b}$$

For a uniform chain, $f(x) \equiv 1$, such a tight-binding chain is conformally critical, with one left- and one right-moving set of fermionic modes near the Fermi energy. The corresponding BCFT spectrum therefore matches that of *one* copy of the chiral fermion (or, equivalently, chiral boson [35]) CFT, whose levels are $1, 1, 2, 3, 5, 7, 11, 15, 22, \ldots$-fold degenerate in each charge sector, while the different charge sectors are offset by $(\Delta q)^2/2$, where $\Delta q$ is the charge deviation from half-filling [36].

We first obtained the ground-state entanglement spectrum of conformal chains [$f(x)$ given by (7, 9a)] of length 256 and 258 with $\Delta = 1/4$, shown in Fig. 2(a,b). As expected, the BCFT spectrum described above is realised in both cases; the different structure of the two plots is due to half-filling occurring at 64 and 64.5 fermions in each half of the chain, respectively. The conformal spectra are extremely accurate: The multiplicity of even the 11th conformal multiplet can easily be read off from the plot, in sharp contrast to a uniform chain of the same length [Fig. 2(d)], where only the first few levels of the tower can be resolved.

We also computed the entanglement spectrum for a 256-site conformal ring [$f(x)$ given by (12)], together with the reflection eigenvalues of the Schmidt states. This is shown in Fig. 2(c): we again obtain very sharp conformal multiplets, which are also distinguished by their alternating reflection eigenvalues in each charge sector, as predicted.

To study the dependence of finite-size corrections on the parameters of the chains, we computed entanglement spectra for a range of system sizes ($8 \leq L\Delta \leq 64$ for $\Delta = 1/2$, 1/4, 1/8, 1/16, 1/32). We analyse these finite-size effects to the entanglement entropy and the structure of the entanglement spectrum in detail in Sec. 3.3. For now, we point out that the length of the chain at fixed $\Delta$ has a strong effect on the conformal multiplets: they are much more clearly resolved in long chains [Fig. 2(e)]. By contrast, changing $\Delta$ even by a factor of 16 at fixed $L\Delta$ has only a modest effect on the entanglement spectrum [Fig. 2(f)]. In fact, the two-parameter scaling can effectively be collapsed by introducing an effective chain length $L_{\text{eff}}$ based on the von Neumann entropy (see Sec. 3.3 and Appendix A), as illustrated in Fig. 2(g). For the system sizes we can access, we find that the deviation of levels from the thermodynamic limit scales as $1/L_{\text{eff}}^2$: This is consistent with earlier results on the scaling

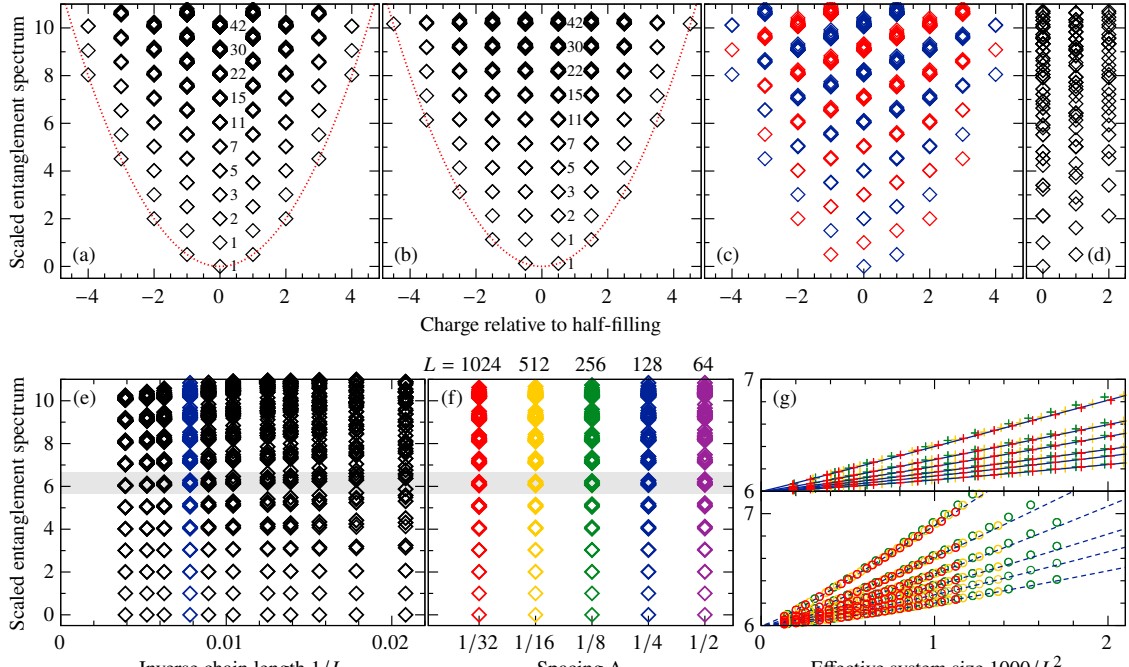

Figure 2: **(a–b)** Ground-state entanglement spectra of conformal (7) free-fermion chains (14) with $\Delta = 1/4$ and length **(a)** $L = 256$, **(b)** $L = 258$. Up to 11 degenerate subspaces with the expected multiplicities $1, 1, 2, 3, 5, 7, 11, \dots$ are clearly seen in each charge sector. The red parabolas indicate the expected $(\Delta q)^2/2$ off-set of entanglement energy between charge sectors. **(c)** Entanglement spectrum of the conformal (12) free-fermion ring for $\Delta = 1/4$ and $L = 256$. Colours indicate mirror-symmetry eigenvalues $+1$ (blue) and $-1$ (red) of the Schmidt vectors, which distinguish subspaces with even and odd momentum. **(d)** Part of the ground-state entanglement spectrum of the free-fermion chain with uniform $J$ and $L = 256$. Unlike the rainbow chains, all but the first few CFT multiplets are too far broadened to distinguish. **(e–f)** Entanglement spectrum in the half-filled sector for several conformal chains (12) with **(e)** fixed $\Delta = 1/4$, **(f)** fixed $L\Delta = 32$ [blue in panel (e)]. **(g)** Schmidt values in the $n = 6$ [11-fold degenerate, grey background in panels (e, f)] conformal level as a function of the effective system size $L_{\text{eff}}$ (18) for conformal chain (top) and ring (bottom) geometries. $\Delta = 1/4, 1/2$ are omitted to reduce clutter. Fits to the form $1/(aL_{\text{eff}}^2 + bL_{\text{eff}}^3)$ are added as guides to the eye. All entanglement cuts are at the middle of the chain. The leading entanglement eigenvalue is shifted to $(\Delta q)^2/2 = 0$ or $1/8$, and the first gap within the half-filling charge sector is scaled to 1.

properties of the entanglement spectrum in the rainbow chain [20, 21], as well as the energy spectrum of CFT critical points on a lattice [37]. However, as we shall see in Sec. 3.3, this scaling does not live up to closer scrutiny of the largest systems.

## 3.2 Majorana fermions (critical Ising chain)

Next, we consider the one-dimensional transverse-field Ising (TFI) chain

$$H = -J \sum_{i=1}^{L-1} f(i + 1/2) \sigma_i^x \sigma_{i+1}^x + h \sum_{i=1}^{L} f(i) \sigma_i^z. \tag{15a}$$

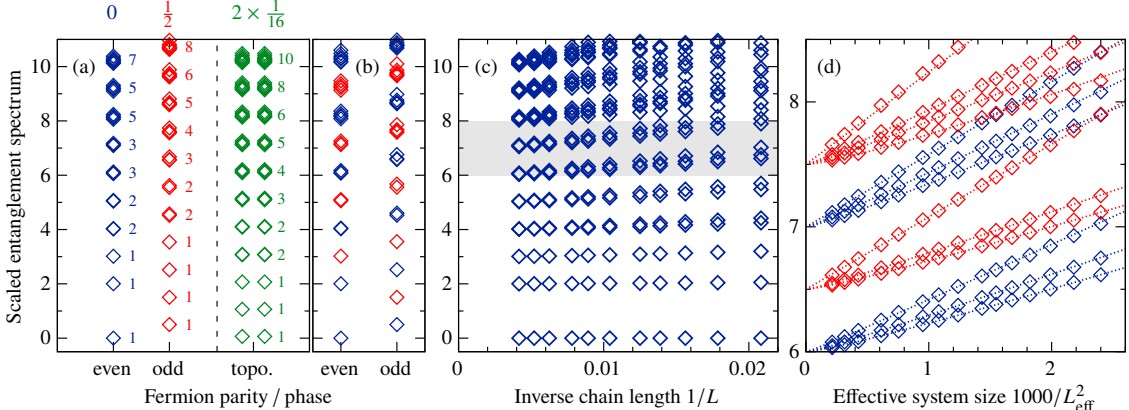

Figure 3: **(a)** Ground-state entanglement spectrum of the conformal (7) TFI chain (15) of length $L = 192$ and $\Delta = 1/4$ in the even and odd fermion-parity sectors (blue and red), and after a boundary deformation to induce the ferromagnetic/SPT phase (green). The number next to each CFT multiplet indicates its multiplicity; the numbers above each spectrum identify the corresponding conformal tower of the Ising CFT: In the paramagnetic/trivial phase, we obtain the identity and fermion sectors; in the ferromagnetic/SPT one, two copies of the twist sector. **(b)** Entanglement spectrum of the conformal (12) TFI ring for $L = 192$ and $\Delta = 1/4$. Colours indicate mirror-symmetry eigenvalues $+1$ (blue) and $-1$ (red) of the Schmidt vectors, which distinguish subspaces with even and odd momentum. **(c)** Entanglement spectrum in the even-parity sector for several conformal chains with fixed $\Delta = 1/4$. **(d)** Schmidt values in the $h + n = 6, 6.5, 7, 7.5$ conformal levels [gray in panel (c)] as a function of effective system size $L_{\text{eff}}$ (18). Fits to the form $1/(aL_{\text{eff}}^2 + bL_{\text{eff}}^3)$ are added as guides to the eye. All entanglement cuts are at the middle of the chain. The leading entanglement eigenvalue is shifted to 0 (1/16) and the first gap scaled to 1/2 (1) in the trivial (topological) chain.

The uniform Hamiltonian $f(x) \equiv 1$ has a transition between a ferromagnetic and a paramagnetic phase at $J = h$, which is described by the two-dimensional Ising CFT. In the entanglement spectrum of the corresponding rainbow-deformed chains, therefore, we expect to recover conformal blocks of the chiral Ising CFT.

To compute the entanglement spectrum of (15a), we map it to a quadratic nearest-neighbour Hamiltonian of Majorana fermions using the Jordan–Wigner transformation (13):

$$H = -J \sum_{i=1}^{L-1} f(i + 1/2) \left( c_i^\dagger - c_i \right) \left( c_{i+1}^\dagger + c_{i+1} \right) + h \sum_{i=1}^{L} f(i) \left( 2c_i^\dagger c_i - 1 \right) \tag{15b}$$

$$= J \sum_{i=1}^{L-1} i f(i + 1/2) \gamma_{2i} \gamma_{2i+1} + h \sum_{i=1}^{L} i f(i) \gamma_{2i-1} \gamma_{2i}, \tag{15c}$$

where we introduce the Majorana operators

$$\gamma_{2i-1} = c_i^\dagger + c_i, \qquad \gamma_{2i} = i \left( c_i^\dagger - c_i \right). \tag{16}$$

For uniform $f(x) \equiv 1$, the Hamiltonian (15c) describes the Kitaev chain [38], in which the transition at $J = h$ becomes one between a topologically trivial ($h > J$) and a symmetry-protected topological (SPT, $h < J$) phase, the latter of which has free Majorana edge modes. We found that wave functions belonging to the topological phase can be induced at the critical point by explicitly decoupling the first and last Majorana modes from the rest of the chain,

allowing us to always work with the critical Hamiltonian in the bulk.[2] In variational optimisation methods (e.g., DMRG), similar gossamer perturbations to drive the critical system into specific phases may be introduced by enforcing different on-site symmetries in the MPS [41].

The ground-state entanglement spectrum of the conformal [$f(x)$ given by (7, 9b)] TFI chain with $J = h$ is shown in Fig. 3(a). This spectrum consists of the conformal blocks (0) and (1/2) of the $\mathcal{M}_3$ minimal model [13], which can be distinguished by the fermion parity of the corresponding Schmidt states: this matches the expected Ising BCFT spectrum with free boundaries [42]. By decoupling the first and last Majorana modes from the Hamiltonian, we obtain an entanglement spectrum that is identical in both parity sectors, both matching the (1/16) conformal block. The perfect twofold degeneracy of the entanglement spectrum, related to the presence of Majorana edge modes, is a hallmark of an SPT phase [2].

The entanglement spectrum in the conformal ring geometry [$f(x)$ given by (12)] also recovers the $(0) \oplus (1/2)$ conformal blocks expected for free boundary conditions [Fig. 3(b)]; as in the complex-fermion case, reflection eigenvalues of the corresponding Schmidt vectors correspond to the parity of the level of the conformal multiplets. Somewhat surprisingly, we were unable to find a deformation of this geometry under which the entanglement spectrum recovers the (1/16) chiral conformal block.

## 3.3 Finite-size scaling

**Entanglement entropy.** As shown in Appendix A, the ground-state entanglement entropy of our models obeys a volume law with a $\Delta$-dependent correction:

$$S_{\mathrm{vN}} \simeq \frac{c}{12}[L\Delta + \ell(\Delta)], \qquad \ell(\Delta) \simeq \begin{cases} -2\log\Delta + \mathrm{const.}\,, & \text{chain geometries,} \\ -4\log\Delta + \mathrm{const.}\,, & \text{ring geometries,} \end{cases} \tag{17}$$

where $c$ is the central charge of the CFT, $c = 1$ and $1/2$ for the tight-binding model (14) and the transverse-field Ising model (15), respectively. To check these predictions, we computed the entanglement spectrum for a range of system sizes ($8 \leq L\Delta \leq 64$ with $\Delta = 1/2$, $1/4$, $1/8$, $1/16$, $1/32$) for both Hamiltonians and all three geometries (7, 8, 12). Von Neumann entropies as a function of system size are plotted in Fig. 4(a, d), respectively: The prediction (17) is borne out perfectly in all cases. It is worth noting that $\ell(\Delta)$ for the TFIM matches $\ell(\Delta/2)$ for the tight-binding chain very closely: This underlines the notion that Hamiltonians with both two-body and one-body terms can be viewed as a discretisation of the continuum CFT Hamiltonian with a doubled number of reference points. To control for the geometry- and $\Delta$-dependence of the entanglement entropy, we will use

$$L_{\mathrm{eff}} = \frac{12}{c}S_{\mathrm{vN}}\,, \tag{18}$$

as our measure of system size from here on.

**Entanglement gap.** We also find analytically (Appendix B) that the entanglement gap $\delta_{\mathrm{ent}}$ within the leading charge sector is inversely proportional to the entanglement entropy for both Hamiltonians:

$$\delta_{\mathrm{ent}} \simeq \frac{c\pi^2}{3S_{\mathrm{vN}}} \simeq \frac{4\pi^2}{L_{\mathrm{eff}}}\,. \tag{19}$$

Our numerical results again match this prediction perfectly, albeit with stronger finite-size (but no visible finite-$\Delta$) effects [Fig. 4(b, e)].

---

[2]We can view this as a boundary CFT with a negative (relative to the bulk) transverse field on the boundary. This boundary condition does not map to any of the standard boundary conditions given by Cardy states [16]; instead, the boundary state is the superposition of the two longitudinal-field Cardy states, $|+\rangle\rangle + |-\rangle\rangle$. This situation is somewhat similar to the case of longitudinal fields in the three-state Potts model, which induce the "new" boundary conditions [39, 40].

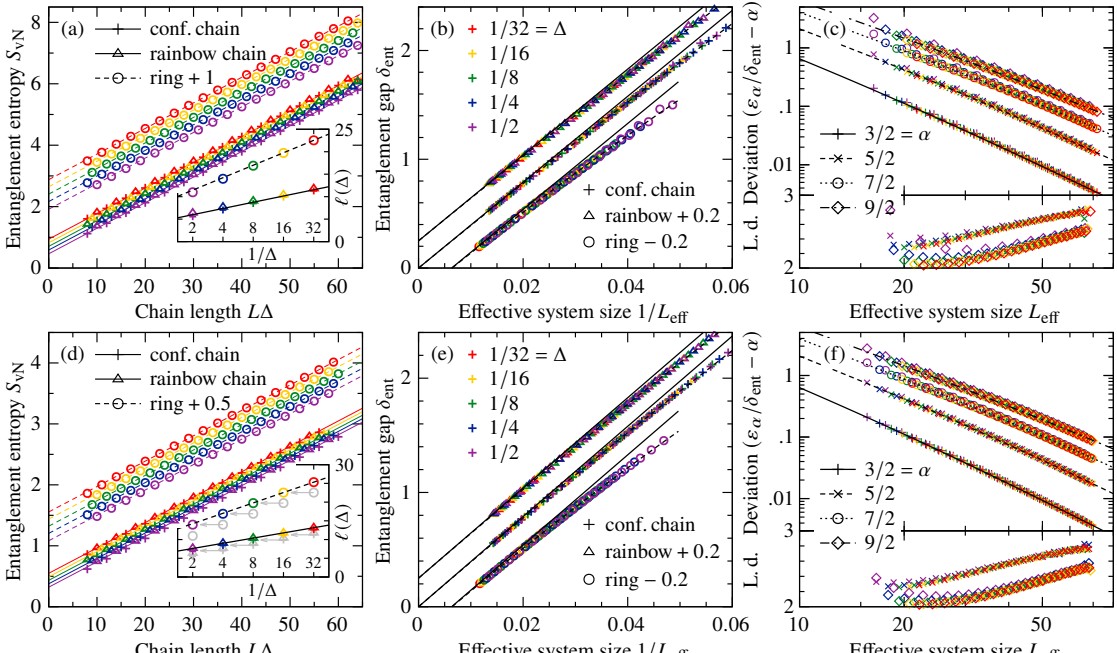

Figure 4: **(a)** Entanglement entropy $S_{vN}$ of the tight-binding model (14) as a function of system size for five different values of $\Delta$ [indicated by colours, see inset and panel (b)]. The data for conformal rings is shifted by $+1$ to improve readability. They obey the volume law (17) (straight lines) to a good approximation. *Inset:* constant volume-law offset $\ell$ as a function of $\Delta$ for the five values of $\Delta$. They follow the expected scaling (17) (black lines). **(b)** Entanglement gap $\delta_{ent}$ in the dominant charge sector as a function of the effective system size $L_{eff} = \frac{12}{c} S_{vN}$. Data for the rainbow-chain and conformal ring geometries are shifted by $\pm 0.2$ to improve readability. For large $L_{eff}$, the inverse proportionality (19) holds (solid lines); with a $1/L_{eff}^2$ correction (dashed lines), it fits for every system size. **(c)** *Top:* Deviation of single-particle entanglement energies $\varepsilon_\alpha$ ($\alpha = 3/2, 5/2, 7/2, 9/2$) from the thermodynamic-limit expectation for rainbow chains. Fits to the form $1/(aL_{eff}^2 + bL_{eff}^3)$ are added as guides to the eye. *Bottom:* Logarithmic derivative of the above for $\alpha = 5/2$ and $9/2$. **(d–f)** The same data for the transverse-field Ising model (15). *Inset of (d):* $\ell(\Delta)$ for the TFI chain very closely matches $\ell(\Delta/2)$ for the tight-binding model (grey).

**Entanglement spectrum.** Since the ground states of both Hamiltonians are fermionic Gaussian states, their reduced density matrices can be written in the form

$$\rho_A = \prod_\alpha \left[ \lambda_\alpha d_\alpha^\dagger d_\alpha + (1 - \lambda_\alpha) d_\alpha d_\alpha^\dagger \right], \tag{20}$$

so Schmidt states are Fock states in the basis of $d$ operators, while the corresponding Schmidt values are products of $\lambda_\alpha$ and $1 - \lambda_\alpha$. For tight-binding Hamiltonians, $\lambda_\alpha$ are the eigenvalues of the correlator matrix $\langle c_i^\dagger c_j \rangle$ restricted to subsystem $A$ [29], while mode creation operators follow from the eigenvectors $v_\alpha$ as $d_\alpha^\dagger = \sum_i (v_\alpha)_i c_i^\dagger$. For Hamiltonians with pairing terms, we need to diagonalise the Nambu correlator matrix instead [31]. Due to Nambu symmetry, we get pairs of eigenvalues $(\lambda_\alpha, \lambda_{\alpha'} = 1 - \lambda_\alpha)$ with mode operators $d_\alpha^\dagger = d_{\alpha'}$: of course, only one from each pair should be included in (20).

In both cases, the entanglement spectrum and its scaling properties are governed by those of the "single-particle entanglement spectrum" $\lambda_\alpha$. It is useful to express these in terms of the

the "single-particle entanglement energies"

$$\varepsilon_\alpha = \log \frac{1 - \lambda_\alpha}{\lambda_\alpha}, \tag{21}$$

which measure the change in entanglement energy upon filling the $d_\alpha^\dagger$ mode in a Schmidt state. Upon the conformal transformations (2, 10), these map onto the single-particle energies of an approximately uniform chain: Near the Fermi level, these have a sinusoidal dispersion that can be expanded as [20]

$$\varepsilon_\alpha \simeq A\frac{\alpha}{L\Delta} - B\frac{\alpha^3}{(L\Delta)^3} + O(L^{-5}), \qquad \alpha = \begin{cases} \pm\frac{1}{2}, \pm\frac{3}{2}, \ldots, & L = 4n, \\ 0, \pm 1, \pm 2, \ldots, & L = 4n + 2. \end{cases} \tag{22}$$

For large systems, higher-order terms of (22) are suppressed in powers of $1/L\Delta$, so the dispersion is well approximated as linear: In this limit, all Schmidt values within a conformal level coincide exactly. Numerically obtained entanglement energies indeed converge closer together with increasing $L$ [Fig. 2(e) and Fig. 3(c)], and $\varepsilon_\alpha/\delta_{\text{ent}}$ converges to $\alpha$ [Fig. 4(c, f)].

Corrections to this limit, however, do not follow (22), which predicts that

$$\frac{\varepsilon_\alpha}{\delta_{\text{ent}}} \equiv \frac{\varepsilon_\alpha}{\varepsilon_1} = \alpha - \frac{B}{A}\frac{\alpha^3 - \alpha}{(L\Delta)^2} + O(L^{-4})$$

approaches $\alpha$ *from below,* the difference scaling as $(L\Delta)^{-2}$. Instead, $\varepsilon_\alpha/\delta_{\text{ent}}$ remains *above* $\alpha$ and converges faster than quadratically [Fig. 4(c)]. The data fit well to $\frac{\varepsilon_\alpha}{\delta_{\text{ent}}} - \alpha \approx 1/(aL_{\text{eff}}^2 + bL_{\text{eff}}^3)$, suggesting an $L_{\text{eff}}^{-3}$ scaling for large systems. This is explained by the structure of the rainbow chain in the middle: For $\Delta \lesssim 1$, $f(x)$ is nearly uniform, corresponding to reference points $\tilde{x}$ in Fig. 1(b) laid out nearly uniformly. Upon the conformal transformation (2), this is mapped onto a chain with linearly tapered couplings on one end, whose energy spectrum has the same features (Appendix A.1). In all, we find that finite-size effects on the structure of the entanglement spectrum have a *cubic* scaling in $L_{\text{eff}}$:

$$\frac{\varepsilon_\alpha}{\delta_{\text{ent}}} - \alpha \simeq +L_{\text{eff}}^{-3}. \tag{23}$$

Accordingly, we expect that deviations of the entanglement spectrum from its expected thermodynamic-limit structure also scale as $L_{\text{eff}}^{-3}$, since these are sums of single-particle entanglement energies $\varepsilon_\alpha$. However, the numerical results in Fig. 2(g) and 3(d) appear more consistent with the naïvely expected quadratic scaling. The reason for the discrepancy is that the cubic scaling only sets in for rather large systems [see the logarithmic derivatives in Fig. 4(c, f)], so for the system sizes that take up most of the figure, the quadratic scaling is a better approximation. In fact, fitting to the form $1/(aL_{\text{eff}}^2 + bL_{\text{eff}}^3)$ proved more accurate than either of the power laws.

# 4 Interacting systems: Numerical renormalisation group

We next turn to Hamiltonians that do not admit a free-fermion representation. It would be natural to obtain the ground-state wave function and its entanglement spectrum by variationally optimising an MPS representation using DMRG. However, we found that this algorithm fails to converge once the dynamical range of Hamiltonian terms [or, equivalently, of $f(x)$] exceeds two or three orders of magnitude; for comparison, the longest chain used in Sec. 3.1 has terms that differ by a factor of $\approx 5 \times 10^{13}$. We believe that this is mostly due to numerical instability: Minimising the local Hamiltonian on weak bonds is sensitive to environments

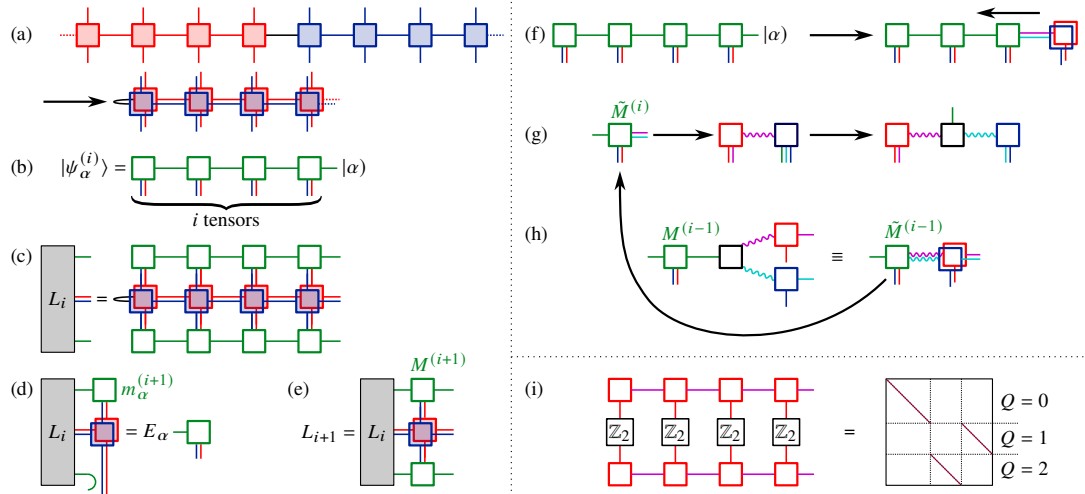

Figure 5: Summary of the numerical renormalisation-group (NRG) algorithm. The MPO representing the Hamiltonian is folded at the strongest bond of the rainbow chain **(a)**, so that sites with equal interaction strengths on the left and the right can be merged. The low-energy eigenstates of the Hamiltonian are likewise encoded in an MPS with doubled physical legs **(b)**; using these, an environment tensor $L_i$ can be defined to capture the low-energy physics of the middle $2i$ sites of the chain **(c)**. The NRG step now consists of diagonalising the effective Hamiltonian **(d)**; the lowest-energy $\chi_{\mathrm{NRG}}$ eigenvectors build up the new MPS tensor $M^{(i+1)}$, which is used to update the environment tensor **(e)**, now describing a segment of length $2(i+1)$. To obtain the entanglement spectrum and build an MPS wave function with the natural ordering of lattice sites, the resulting MPS must be "unzipped" **(f)**. This is done by splitting isometries involving the left and right physical and virtual legs off $\tilde{M}^{(i)}$ **(g)** and fusing the remainder into $M^{(i-1)}$ **(h)**, which yields $\tilde{M}^{(i-1)}$ for the next step. **(i)** Schmidt states of the three-state Potts model with trivial $\mathbb{Z}_3$ charge are eigenstates of the $\mathbb{Z}_2$ parity operator with eigenvalue $\pm 1$, distinguishing the $0^{\pm}$ irreps of $S_3$; the other two charge sectors map onto each other.

that include much larger Hamiltonian terms, so a tiny improvement on their energy contribution may outweigh optimising the weaker bonds altogether. This effect is exacerbated by the long-range correlations of rainbow chains [cf. Fig. 1(c)].

Instead, we developed a numerical renormalisation-group (NRG) algorithm inspired by Wilson's approach to the Kondo problem [23]. Here, the metallic bath was represented using a semi-infinite tight-binding chain (known as the *Wilson chain*) with exponentially decaying hopping terms, which allows for extracting the low-energy physics by iterative diagonalisation: Lowest-energy eigenstates of the first $(i + 1)$ sites of the chain only have considerable overlap with the lowest-energy eigenstates of the first $i$ sites, since the relatively weak coupling terms to the last site do not introduce significant perturbations from higher energies. Therefore, the lowest-energy eigenstates at every chain length can be approximated by iteratively diagonalising and restricting the Hamiltonian in the next step to the lowest few eigenstates.

This process can naturally be written as the iterative construction of an MPS representation [Fig. 5(b)] of the lowest eigenstates [43, 44]. The action of the Hamiltonian [represented as a matrix-product operator (MPO)] on the lowest eigenstates of a chain of length $i$ can be compressed into an environment tensor [Fig. 5(c)], which yields the effective Hamiltonian for step $(i + 1)$ [Fig. 5(d)]. The new MPS tensor is made up of the lowest-energy eigenvectors of this effective Hamiltonian; the new environment tensor for $(i + 1)$ sites can be built from this tensor and the previous environment [Fig. 5(e)].

This NRG protocol is popular for solving the Kondo problem and the Anderson impurity problem in dynamical mean-field theory [24], but has seen little use outside of this context. Nevertheless, it is a convenient choice for solving rainbow chains (8), whose Hamiltonian terms also decay exponentially: The approximation of iterative diagonalisation and discarding high-energy states remains valid even if the Hamiltonian is not a free-fermion one. Furthermore, if the rainbow chain is tuned to a critical point, we expect the NRG procedure to converge to a corresponding fixed point, which can be verified numerically.

The only change to the algorithm outlined above is that the Hamiltonian terms of equal strength on the two sides of the chain must be handled simultaneously, requiring us to fold the MPO Hamiltonian at the central bond of the rainbow chain [Fig. 5(a)].[3] Accordingly, the low-energy states are represented by MPS whose tensors have *two* physical legs, corresponding to one site each in the left and right halves of the chain [Fig. 5(b)]. This in fact matches the entanglement structure of the expected rainbow ground state, which, in the limit $\alpha \to 0$, consists of singlet dimers precisely between the fused sites [Fig. 1(c)]: such a state can be represented as an MPS of bond dimension 1. More generally, the folded representation naturally allows for volume-law entanglement, since the left and right halves of the chain are not coupled through a single MPS bond, but rather by *all the tensors*. This means that, in principle, we can obtain the ground state of a rainbow chain of arbitrary length and dynamical range of parameters, even outperforming the free-fermion method, which relies on numerically diagonalising the full single-particle Hamiltonian.

**"Unzipping" folded MPS into standard MPS.** However, extracting the entanglement spectrum from such a representation is not straightforward, precisely because the half-chain entanglement is spread out across all tensors of the MPS, rather than a single central bond. We dealt with this issue by converting the folded MPS into a standard one by separating each tensor into two tensors that carry the left and right physical legs, respectively: We work from the last tensor (carrying the first and last physical sites) towards the first (representing the centre of the chain) iteratively, "unzipping" the folded MPS [Fig. 5(f)].

At each iteration, the algorithm splits off two isometries from the folded-MPS tensor [Fig. 5(g)]: one carries the left physical (red) and incoming virtual[4] (magenta) legs, the other the right (blue and cyan) legs. This can be done using either QR or singular-value decompositions (SVD); we prefer the latter, as it gives a more controlled way to truncate the decomposition to a fixed bond dimension. The new legs generated by the decompositions (wiggly magenta and cyan) act as outgoing virtual legs. Finally, the remaining tensor (black) is fused to the next tensor to be unzipped [Fig. 5(h)], and the whole process repeated until all folded-MPS tensors are decomposed. The last tensor, representing the two central sites, has no virtual leg going towards the middle (green), so it can be decomposed with a single SVD. Since we have split off isometries at every step, the now fully unzipped MPS is in mixed canonical form, so the entanglement spectrum can be read off directly from the singular values in the last step.

## 4.1 Example: The three-state Potts model

We demonstrate the algorithms outlined above on the ferromagnetic three-state Potts model

$$H = -J \sum_{i=1}^{L-1} f(i+1/2)(X_i X_{i+1}^\dagger + X_i^\dagger X_{i+1}) - h \sum_{i=1}^{L} f(i)(Z_i + Z_i^\dagger) \tag{24}$$
$$- h_L(X_1 + X_1^\dagger) - h_R(X_L + X_L^\dagger),$$

---

[3]This approach is identical to the standard NRG treatment of spinful baths, represented by two Wilson chains that only interact via the impurity site [23].

[4]For the first step, virtual legs can either be omitted or trivial ones added for convenience.

Table 1: Irreducible representations (irreps) of the $S_3$ on-site symmetry group of the three-state Potts model and their decompositions into irreps of the $\mathbb{Z}_3$ subgroup preserved by $\hat{Z}$ and the $\mathbb{Z}_2$ subgroup preserved by $\hat{X} + \hat{X}^\dagger$, labelled respectively with a $\mathbb{Z}_3$ charge and a parity.

| Label | dimension | $\mathbb{Z}_3$ | $\mathbb{Z}_2$ |
|:-----:|:---------:|:--------------:|:--------------:|
| $0^+$ | 1 | 0 | $+$ |
| $0^-$ | 1 | 0 | $-$ |
| $E$ | 2 | $1 \oplus 2$ | $+ \oplus -$ |

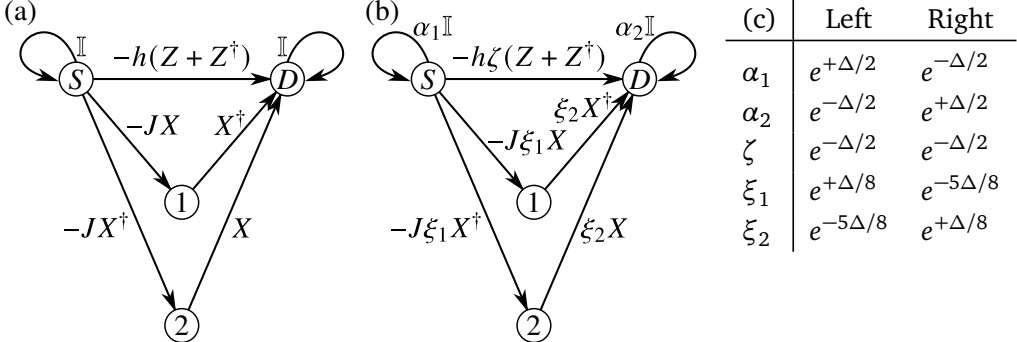

Figure 6: Graphs of the finite-state automata that represent the three-state Potts Hamiltonian (24) **(a)** with uniform couplings $f(x) \equiv 1$ and **(b)** as a rainbow chain (26). **(c)** Values of the coefficients in (b) in the left and right halves of the rainbow chain.

where $X$ and $Z$ are $3 \times 3$ matrices acting on each site, satisfying $XZ = \omega ZX$ ($\omega = e^{2\pi i/3}$ is the third root of unity), e.g.,

$$X = \begin{pmatrix} 1 & 0 & 0 \\ 0 & \omega & 0 \\ 0 & 0 & \omega^2 \end{pmatrix}, \qquad Z = \begin{pmatrix} 0 & 1 & 0 \\ 0 & 0 & 1 \\ 1 & 0 & 0 \end{pmatrix}. \tag{25}$$

In the absence of the boundary fields $h_{L,R}$, Eq. (24) is symmetric under simultaneous permutations of the basis states in (25) on all sites. These on-site symmetries form a non-Abelian $S_3$ group: eigenstates and Schmidt states can be labelled with irreducible representations (irreps) of this group, which are listed in Table 1. In numerical simulations with TeNPy [45], we enforced the largest Abelian subgroup of $S_3$, the $\mathbb{Z}_3$ group of cyclic permutations. Irreps of this group are labelled by eigenvalues of $Z$, so the Hamiltonian can be written in the irrep basis by swapping the definitions of $X$ and $Z$ in (25). We shall also use $Q = 0, 1, 2 \pmod 3$ to denote the $Z$ eigenvalues $1, \omega$, and $\omega^2$. The boundary fields $h_{L,R}$, on the other hand, break this $\mathbb{Z}_3$ symmetry, leaving only the $\mathbb{Z}_2 < S_3$ group generated by swapping the $\omega, \omega^2$ eigenvalues of $X$ (or, in fact, of $Z$). The correspondence between the irreps of $S_3$ and of these groups is also listed in Table 1.

**Representing the rainbow chain as a matrix-product operator.** Local Hamiltonians can readily be converted to MPO form by first representing them as finite-state automata [46,47]. These can be thought of as a directed graph, where each edge is assigned an operator acting on a single site: Each term in the Hamiltonian for an $L$-site system is the product of these operators along a path of length $L$ between designated "source" and "drain" nodes $S$ and $D$. For example, the simplest finite-state automaton that represents (24) for a uniform chain ($f \equiv 1$) is shown

in Fig. 6(a). The edge between the source and the drain generates the $-h(Z + Z^\dagger)$ terms, while the paths $S \to 1 \to D$ and $S \to 2 \to D$ generate the nearest-neighbour $-JXX^\dagger$ and $-JX^\dagger X$ terms (a path going from $S$ to 1 or 2 must continue to $D$ on the next step, so the operators are bound to act on neighbouring sites). Finally, the self-edges $S \to S$ and $D \to D$ make sure that the automaton can generate all terms of the form $I \ldots I Z I \ldots I$, $I \ldots I X X^\dagger I \ldots I$, and $I \ldots I X^\dagger X I \ldots I$ for a chain of arbitrary length. The nodes of such a representation can then be viewed as indices on the virtual leg of the MPO, while the operator assigned to each edge $i \to j$ represents the block $W(i, j)$ in the MPO tensor. Each path $S \to i \to j \to \cdots \to k \to D$ corresponds to the product $W(S, i)W(i, j) \cdots W(k, D)$: the matrix product $W(S, \cdot)W \cdots W(\cdot, D)$ is indeed the sum of all products of this form.

To represent the rainbow chain $[f(x)$ given by (8, 9b)$]$ instead, we need the MPO to scale each of the Hamiltonian term according to its position in the chain, ideally using position-independent tensors (at least in each half of the chain). In particular, to keep fixed-point NRG energies invariant, we need to scale the Hamiltonian such that the coefficients of the outermost terms remain independent of chain length. More specifically, we set the coefficient of $(Z_1 + Z_1^\dagger)$ to be $f(1) = 1$, whence

$$f(x) = f(L - x) = e^{\Delta(x-1)} \qquad (x \leq L/2), \tag{26a}$$

$$f\left(\frac{L+1}{2}\right) = e^{\Delta(L/2 - 3/4)}. \tag{26b}$$

These exponentially growing coefficients can readily be implemented by changing the operator on the self-edges $S \to S$ and $D \to D$ to a multiple of the identity Fig. 6(b): the coefficient of each Hamiltonian term is the product of these factors, hence they are exponential in the number of factors, i.e., the length of the chain. In particular, we find that scaling the $S \to S$ and $D \to D$ edges by $e^{+\Delta/2}$ and $e^{-\Delta/2}$ on the left half of the chain and vice versa on the right half provides the correct scaling: A sequence of $x \leq L/2$ $S \to S$ and $L - x$ $D \to D$ self-edges accumulate a total factor of

$$e^{x\Delta/2} e^{-(L/2 - x)\Delta/2} e^{L/2 \times \Delta/2} = e^{x\Delta},$$

a scaling with distance that matches (26). We find the same prefactor for $L - x$ $S \to S$ and $x$ $D \to D$ self-edges.

In the notation of Fig. 6(b), the coefficient of $(Z_1 + Z_1^\dagger)$ and $(Z_N + Z_N^\dagger)$ is

$$f(1) = f(L) = \zeta e^{-\Delta(L/2 - 1)/2} e^{+\Delta(L/2)/2} = \zeta e^{\Delta/2};$$

to match (26a), we have to set $\zeta = e^{-\Delta/2}$. Likewise, the coefficients of $X_1 X_2^\dagger$ and $X_{L-1} X_L^\dagger$ are

$$f(3/2) = \xi_1^L \xi_2^L e^{-\Delta(L/2 - 2)/2} e^{+\Delta(L/2)/2} = \xi_1^L \xi_2^L e^{\Delta},$$

$$f(L - 1/2) = \xi_1^R \xi_2^R e^{\Delta},$$

where we allow for $\xi_{1,2}$ to be different in the two halves of the chain. From (26a), both of these should be $e^{\Delta/2}$, hence $\xi_1^L \xi_2^L = \xi_1^R \xi_2^R = e^{-\Delta/2}$. Finally, the coefficient of $X_{L/2} X_{L/2+1}^\dagger$ is

$$f\left(\frac{L+1}{2}\right) = e^{\Delta/2(L/2 - 1)} \xi_1^L \xi_2^R e^{\Delta/2(L/2 - 1)} = \xi_1^L \xi_2^R e^{\Delta(L/2 - 1)};$$

comparing with (26b) sets $\xi_1^L \xi_2^R = e^{\Delta/4}$. It is easy to verify that the coefficients listed in Fig. 6(c) satisfy all of these requirements.

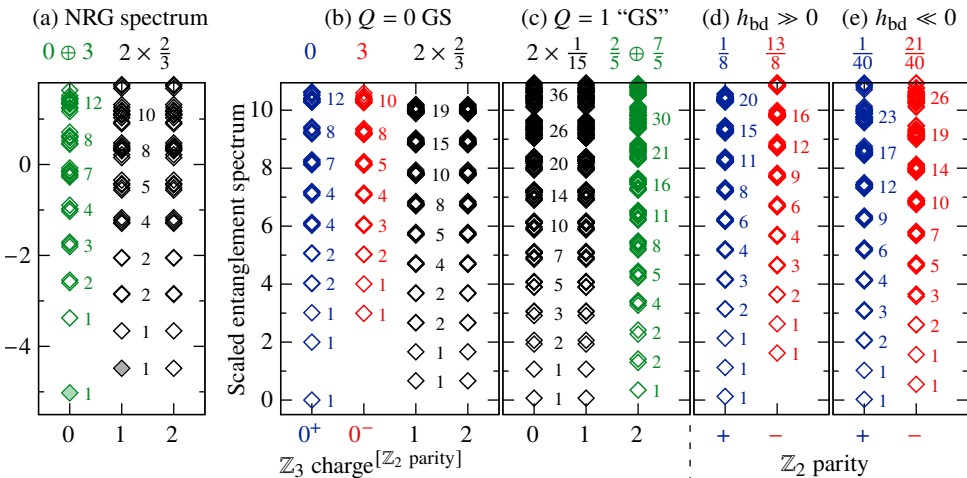

Figure 7: **(a)** Low-energy spectrum of the critical ferromagnetic 3-state Potts model
on a rainbow chain (26), obtained from the numerical renormalisation group (NRG).
**(b–c)** Entanglement spectrum of the ground state **(b)** and the first excited state in
symmetry sector $E$ **(c)**, marked with filled symbols in panel (a). Schmidt vectors
are labelled with their eigenvalue under $\mathbb{Z}_3 < S_3$, as well as their $\mathbb{Z}_2$ parity where
that symmetry is preserved. **(d–e)** Entanglement spectrum of the ground state upon
applying strong positive **(d)** and negative **(e)** boundary fields $h_L, h_R$. Schmidt vectors
are labelled with their parity under the $\mathbb{Z}_2 < S_3$ symmetry preserved by the boundary
fields. The number next to each CFT multiplet indicates its multiplicity; the numbers
above each spectrum identify the corresponding conformal tower of the $\mathcal{M}_5$ minimal
model. Unless otherwise stated, all simulations use $L = 256$, $\Delta = 1/4$, $\chi_{\mathrm{NRG}} = 400$,
$\chi_{\mathrm{unzip}} = 800$. $h/J$ is tuned slightly away from 1 to cancel out the effects of truncating
the low-energy Hilbert space, cf. Sec. 4.2 and Fig. 8. Entanglement spectra are scaled
by the first gap in the leading symmetry sector, and the lowest level is shifted to match
the scaling dimensions of the corresponding primary fields of $\mathcal{M}_5$.

**Fixed-point energy spectrum.** The uniform Potts model ($f(x) \equiv 1$) has a self-dual critical
point at $J = h$, described by the $\mathcal{M}_5$ conformal minimal model [48]. Therefore, we performed
NRG calculations for a rainbow chain [$f(x)$ given by (8, 9b)] at $J \approx h$ (cf. Sec. 4.2), where
we expect that both energy and entanglement spectra are described by BCFT spectra of this
minimal model [16, 42]. Indeed, for sufficiently high $\Delta$ and a sufficiently large number $\chi_{\mathrm{NRG}}$
of low-energy states kept, the NRG procedure converges to a fixed-point spectrum, shown in
Fig. 7(a): The spectrum consists of the conformal block $(0) \oplus (3)$ (in the $0^{\pm}$ irreps of $S_3$) and
two copies of the block $(2/3)$ (in the two-dimensional $E$ irrep), as expected for free boundaries.

**Ground-state entanglement spectrum (free boundaries).** We next obtained the entangle-
ment spectrum of the ground state using the unzipping algorithm in Fig. 5(f–h) [Fig. 7(b)].
We find the same structure as in the energy spectrum, as expected. Since the ground state is
symmetric under the full $S_3$ group, the irreps $0^+$ and $0^-$ can also be distinguished by comput-
ing the eigenvalues of the Schmidt vectors under the $\mathbb{Z}_2$ group generator, as shown in Fig. 5(i).
We find that the (0) and (3) conformal blocks appear in separate irreps: This is a remarkable
difference to the partition function of the boundaryless three-state Potts model [49,50], where
these two blocks are bound together by offdiagonal terms.

**Excited-state entanglement spectrum.** We also computed the entanglement spectrum of the lowest-energy state in the $\mathbb{Z}_3$ symmetry sector $Q = 1$ (i.e., $\prod Z = \omega$), shown in Fig. 7(c): It consists of the conformal blocks $(2/5) \oplus (7/5)$ ($Q = 2$ sector) and two copies of the block $(1/15)$ ($Q = 0, 1$ sectors). This spectrum is consistent with the so-called "new" boundary conditions [39] of the Potts model, which are also induced by applying *transverse* $(Z + Z^\dagger)$ boundary fields [40].

In fact, the entanglement spectra of excited states do not map straightforwardly on a BCFT spectrum. Each (energy) eigenstate $|\psi_\mathcal{O}\rangle$ of a CFT on a finite segment corresponds to a boundary(-changing) operator $\mathcal{O}$ compatible with the two boundary conditions: In $z$-space in Fig. 1(a,b), $\mathcal{O}$ is inserted at $-i\infty$, while $\langle\psi_\mathcal{O}|$ is generated by inserting $\mathcal{O}^\dagger$ at $+i\infty$. Upon the conformal transformation (2), these are mapped to $w = \pi/2$ and $w = 3\pi/2$, respectively [black X in Fig. 1(a,b)]. If the boundary condition on both sides is the same, the operator $\mathcal{O}$ corresponding to the ground state is the identity, so the entanglement spectrum can be viewed as a standard BCFT spectrum. By contrast, for excited states (or if the boundary conditions on the two sides are different), $\mathcal{O}$ is nontrivial, so the entanglement spectrum is no longer a simple BCFT spectrum. Nevertheless, the fact that the spectrum is still organised into conformal towers strongly suggests that the boundary state that describes this more complex boundary configuration on an annulus has a similar structure to standard Cardy states [16]: In future work, it will be interesting to understand these states in more detail and develop predictions for the conformal towers generated in the resulting generalised BCFTs.

**Ground-state entanglement spectrum (fixed boundaries).** Finally, we consider the ground-state entanglement spectrum of (24) in the presence of strong boundary fields $h_{L,R}$. (In practice, we set spins 1 and $L$ to an eigenstate of $\hat{X}$ and modify the Hamiltonian for the last NRG step, spins 2 and $L-1$, to include the terminal two-site terms as well.) Upon the conformal mapping (2), this corresponds to a finite-temperature BCFT with free boundary conditions on the side of the entanglement cut (where the Hamiltonian was not changed) and fixed ones on the other side. The spectrum for this setup was predicted [16, 42] to contain the conformal blocks $(1/8) \oplus (13/8)$ and $(1/40) \oplus (21/40)$ in the limit of strong positive and negative boundary fields, respectively, even though these do not appear in the spectrum of the boundaryless critical Potts model [49, 50]. We indeed recover these in our simulations, see Fig. 7(d,e); the two blocks are readily distinguished by their $\mathbb{Z}_2$ symmetry eigenvalues.

## 4.2 Parameter dependence

**Finite-$\chi_{\mathrm{NRG}}$ effects on NRG stability.** Keeping a limited number $\chi_{\mathrm{NRG}}$ of lowest-energy states in NRG (that is, constructing a folded MPS representation with bond dimension capped at $\chi_{\mathrm{NRG}}$) perturbs the ideal RG flow by truncating the overlap of the ground state with higher-energy eigenstates of shorter segments. As shown in Fig. 8(a), this is not a relevant perturbation: the critical spectrum in Fig. 7(a) is stable for a large span of NRG steps. However, as the critical point is an unstable RG fixed point, these perturbations are magnified over time and eventually drive the flow to one of the two stable fixed points, corresponding to a noninteracting paramagnet ($J \to 0$) and a nondynamical Potts ferromagnet ($h \to 0$), the spectra of which are shown in Fig. 8(b). The significance of this effect also depends on $\Delta$, which controls the separation of scales between NRG steps: For large $\Delta$, high-energy states have a smaller effect on subsequent steps, so a smaller $\chi_{\mathrm{NRG}}$ keeps the NRG procedure stable for more steps, as illustrated in Fig. 8(c).

The perturbation due to the finite $\chi_{\mathrm{NRG}}$ can readily be cancelled by tuning $g \equiv h/J$ slightly away from 1. We found the optimal value by applying the secant method to the ground-state energy evolution away from the unstable fixed point, which is approximately linear in $g$ near the fixed point. With standard floating-point numerics, the renormalised critical point can

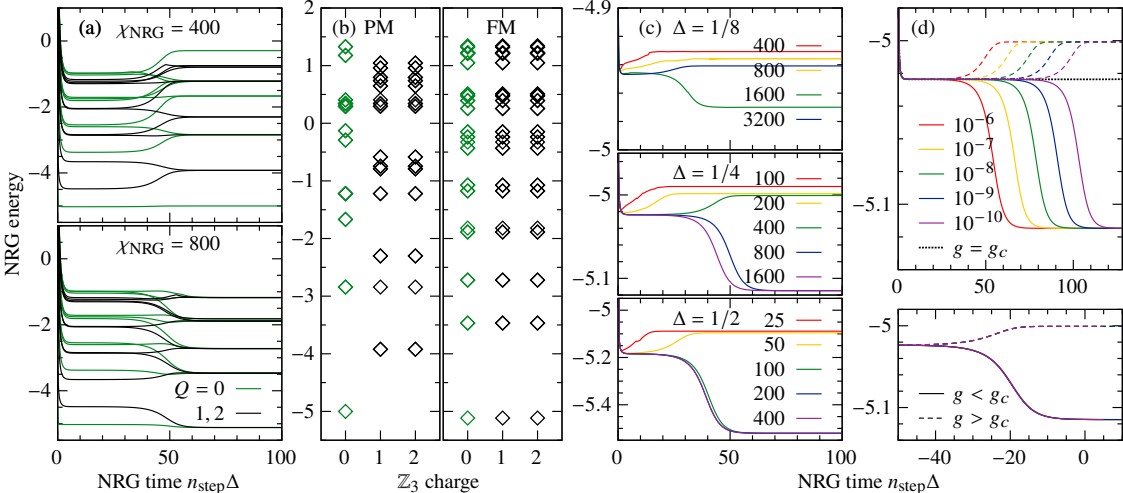

Figure 8: **(a)** Evolution of the lowest 10 NRG eigenstates in each $\mathbb{Z}_3$ charge sector for $\Delta = 1/4$ and $\chi_{\mathrm{NRG}} = 400$ (top) and 800 (bottom), diverging to the paramagnetic and ferromagnetic fixed points, respectively. **(b)** NRG spectrum at the final iteration of the two trajectories. For $\chi_{\mathrm{NRG}} = 800$, the ground state becomes degenerate across the three charge sectors, indicating spontaneous symmetry breaking. **(c)** Evolution of the NRG ground state for three values of $\Delta$ and several values of $\chi_{\mathrm{NRG}}$. Increasing the bond dimension keeps the critical fixed point stable for longer, up to some value of $\chi_{\mathrm{NRG}}$. Achieving the same degree of stability requires substantially higher $\chi_{\mathrm{NRG}}$ for smaller values of $\Delta$. **(d)** Evolution of the NRG ground state for $\Delta = 1/4$, $\chi_{\mathrm{NRG}} = 400$ near the renormalised critical point $g_c = 0.99999645690947$. The profile of the divergence towards the stable fixed points is identical for different values of $g - g_c$, and can be collapsed perfectly by shifting the time axis by $(16/3) \log |g - g_c|$ (bottom).

be computed to about 14 significant figures before floating-point errors wash out any further improvement. For $\Delta = 1/4$, $\chi_{\mathrm{NRG}} = 400$, we found $g_c = 0.99999645690947$; this value is used for Fig. 7.

The effect of small deviations away from the renormalised $g_c$ is shown in Fig. 8(d). We find that the critical fixed point is destabilised after NRG time $\tau \equiv n_{\mathrm{step}} \Delta \simeq \nu_{\mathrm{eff}} \log |g - g_c|$. In fact, the evolution of the NRG spectrum from the critical to the stable fixed points for different $g$ can perfectly be collapsed by shifting them by $\nu_{\mathrm{eff}} \log |g - g_c|$. Since each NRG step increases the effective system size by a factor of $e^\Delta$, we can view this as an effective correlation length scaling as $\xi \sim e^\tau \sim |g - g_c|^{\nu_{\mathrm{eff}}}$, with the data at different values of $g$ obeying standard finite-size scaling. Furthermore, near the critical fixed point, deviations grow exponentially, controlled again by $\nu_{\mathrm{eff}}$: at the $n$th step, $\varepsilon(n) - \varepsilon_0 \propto \exp(n\Delta / \nu_{\mathrm{eff}})$, again consistent with the interpretation of $\nu_{\mathrm{eff}}$ as a correlation length exponent. The numerically extracted $\nu_{\mathrm{eff}}$ depends somewhat on $\chi_{\mathrm{NRG}}$ and $\Delta$; however, for all the curves shown in Fig. 8(c), it is between 5.3 and 5.5,[5] suggesting a universal origin. Surprisingly, however, this value is vastly different to the known correlation length exponent of the three-state Potts model, $\nu = 5/6$.

To better understand this discrepancy, we also performed NRG simulations of the transverse-field Ising model, discussed in more detail in Appendix C. Instead of exponential divergence with the expected $\nu = 1$, we found that deviations from the critical fixed point grow linearly with $n_{\mathrm{step}}$, hinting at $1/\nu_{\mathrm{eff}} = 0$ (with log-corrections). We note that in both cases, we have $\nu_{\mathrm{eff}} = 1/(1 - h_\varepsilon)$ instead of the standard $\nu = 1/(2 - h_\varepsilon)$ to a good approximation, where $h_\varepsilon$ is the (bulk) scaling dimension of the most relevant perturbation that can be added to the

---

[5]For the particular case of $\Delta = 1/4$, $\chi_{\mathrm{NRG}} = 400$ in Fig. 8(d), $\nu_{\mathrm{eff}} = 16/3$ to a good approximation.

Hamiltonian: Namely, for the Potts model (TFIM), $h_\varepsilon = 4/5\ (1)$, whence $\nu_{eff} = 5\ (\infty)$ instead of $\nu = 5/6\ (1)$. Checking this hypothesis for other theories and understanding the origin of the discrepancy will be an interesting direction for future work.

**Finite-$\chi$ effects on the unzipping algorithm.** From here on, we consider the NRG wave function obtained for $\Delta = 1/4$, $\chi_{NRG} = 400$ at the renormalised critical point $g_c$, which we take to be a faithful representation of the ground state of the critical three-state Potts rainbow chain. However, to extract such quantities as the von Neumann entropy and the entanglement spectrum, the folded MPS representation needs to be unzipped into a regular MPS using the algorithm in Fig. 5(f–h), which is controlled by a new hyperparameter, the bond dimension $\chi_{(unzip)}$ of this new MPS.

In Fig. 9(a), we show the von Neumann entropy obtained for a range of different system sizes and values of $\chi$. The entropy of the exact wave function is expected to increase according to the volume law $S \simeq \frac{c}{12}L\Delta$ [cf. (17)]. However, this is capped by the MPS representation, which can only capture $S \leq \log \chi$. The numerical results are consistent with this picture: Before saturation, the correction $\ell(\Delta)$ to the effective chain length matches exactly that for the TFIM (15), highlighting its geometrical origin. The saturation value of $S_{vN}$, however, differs from $\log \chi$ by a constant offset, indicating some surviving structure even in this limit.

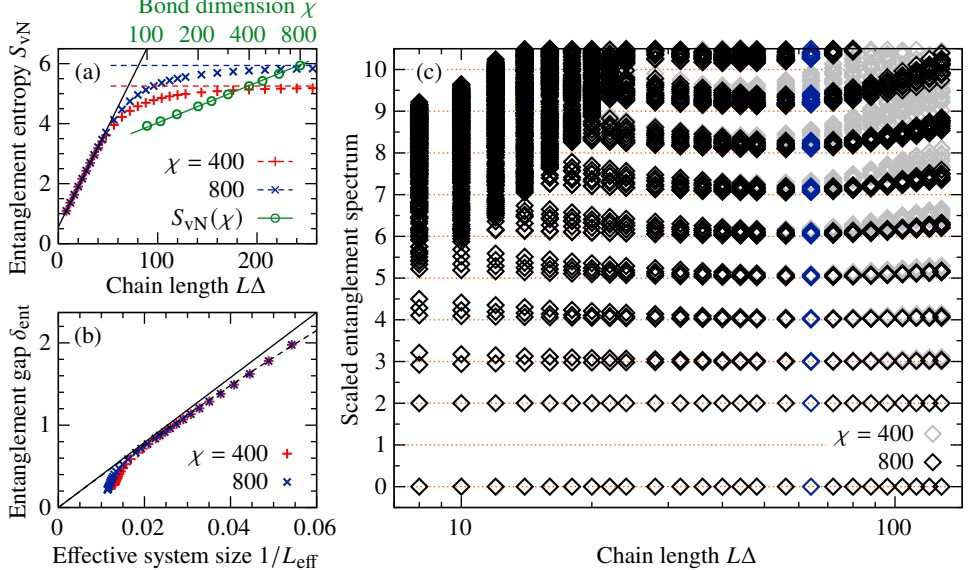

Figure 9: **(a)** Ground-state entanglement entropy of the critical three-state Potts model as a function of system size, obtained using the unzipping algorithm of Fig. 5(f–h) with MPS bond dimension $\chi = 400$ (red) and 800 (blue). They obey the volume law with $\ell(\Delta)$ taken from the transverse-field Ising model (black line) up until saturation due to finite $\chi$ (dashed lines). The saturated values are plotted as a function of $\chi$ (green): they fit well to $S_{sat} = \log \chi - $const. (green line). **(b)** Entanglement gap as a function of effective system size $L_{eff} = \frac{12}{c}S_{vN}$. They follow (19) (black line, quadratic correction in dashes) up to saturation, where $\delta_{ent}$ falls below its expected magnitude. **(c)** Entanglement spectrum as a function of system size obtained with $\chi = 400$ (grey) and 800 (black). Below saturation, they converge towards the expected thermodynamic limit, with little $\chi$-dependence; above it, they diverge again, more strongly for smaller $\chi$. Fig. 7 uses $L = 256$ (blue symbols), where the multiplets are the narrowest.

Before saturation, the entanglement gap $\delta_{\text{ent}}$ (defined as half the gap between the lowest CFT levels in the identity sector, $h + n = 0, 2$) also follows the scaling (19) found previously for free-fermion systems [Fig. 9(b)]. However, upon saturation, it deviates downwards from this scaling. This is to be expected, since the entanglement entropy represented by an MPS of fixed bond dimension is maximised if all Schmidt values are equal, that is, if the entanglement gap vanishes: As the MPS truncation algorithm tries to preserve as much of the entanglement entropy as possible, it pushes towards such a result.

Finally, we plot the rescaled entanglement spectrum in the zero-charge sector in Fig. 9(c). Before the saturation of entanglement entropy, the dependence of this spectrum on the bond dimension is negligible, and the spread of each conformal multiplet decreases as the system size increases, presumably with the same power laws as in the free-fermion case. By contrast, as the entanglement entropy saturates, the multiplets start to broaden again. The extent of this broadening strongly depends on the bond dimension $\chi$: Generally, a larger $\chi$ leads to less broadening, but the $\chi$-dependence is quite erratic and we were unable to extract any scaling law. These artefacts, caused by the saturation of entanglement entropy and the attendant deviation of the unzipped MPS from the real wave function, make the physical results less reliable. Therefore, it is recommended to set the system size and $\chi$ according to

$$S \simeq \frac{c}{12} L\Delta \lesssim \log \chi \quad \implies \quad \chi \gtrsim e^{cL\Delta/12}, \qquad L \lesssim \frac{12 \log \chi}{c\Delta}, \tag{27}$$

to avoid saturation and thus obtain a faithful representation of the actual wave function.

## 5 Conclusion

In this work, we have developed a method to obtain detailed spectral information about conformal field theories using entanglement spectra of inhomogeneous spin chains. By exploiting the conformal mapping between reduced and thermal density matrices [14], we designed Hamiltonians whose entanglement spectra match the energy spectrum of long critical spin chains, thereby removing the obstacle of logarithmic finite-size scaling [13, 15]. These Hamiltonians are still short-range interacting (up to nearest neighbours in our examples), but the magnitude of their terms decays exponentially going out from the middle, similar to the so-called rainbow chains [19–22].

We first demonstrated our method on free-fermion Hamiltonians, where we could obtain entanglement spectra from single-particle calculations [27]. Indeed, we found that entanglement spectra match those of the expected free-fermion and Ising boundary CFTs, with up to 11 levels of conformal blocks clearly resolved in each symmetry sector: This is a significant improvement over uniform chains and the edge spectra of narrow cylinders, where entanglement spectra can often only resolve the first few states in these towers.

We also proposed a Wilsonian numerical renormalisation group (NRG) algorithm for obtaining low-energy eigenstates for rainbow chains that represent interacting (non-free-fermion) Hamiltonians. In contrast to earlier work [43], we find that this algorithm clearly outperforms such alternatives as DMRG for the problem at hand; most importantly, it can handle chains with an extremely wide dynamical range of Hamiltonian terms, as they are optimised hierarchically. Furthermore, the wave function is returned as a folded MPS [Fig. 5(b)], which is well-suited to the volume-law entanglement of the rainbow ground state. We benchmarked the NRG algorithm on the three-state Potts model: By computing ground- and excited-state entanglement spectra for different boundary conditions, we were able to obtain the spectrum of every conformal tower of the $\mathcal{M}_5$ minimal model with excellent accuracy and detail.

As it makes no strong assumptions about the structure of the underlying Hamiltonian, our NRG approach is in principle applicable to any conformal quantum critical point in $(1 + 1)$

dimensions. This allows us to identify the underlying CFTs through access to their full spectra, a major improvement over the entanglement entropy, which only recovers the central charge. In future work, it will be interesting to explore how further properties of the CFT (e.g., the list of primary fields, their scaling dimensions, fusion rules, and OPE coefficients) may be extracted from wave functions or NRG fixed-point tensors: Since wave functions in different symmetry sectors and for different boundary conditions can be extracted from the same NRG flow, all this CFT information must be encoded directly in the NRG flow and in the fixed-point tensor in particular.

A systematic method of extracting all conformal towers is particularly important for characterising unknown CFTs, for which the entanglement spectra of low-energy excited states are a promising resource. In this paper, we only considered the first excited states of the transverse-field Ising model and the three-state Potts model with free boundary conditions: Both of these turn out to match the BCFT partition function with free (at the entanglement cut) and fixed *transverse*-field boundary conditions. Excited-state entanglement spectra are not generated by standard Cardy boundary states and, to the best of our knowledge, have not been studied in detail: In addition to understanding the power and limitations of the numerical method, describing these complex boundaries will yield new insights to boundary and defect CFTs. A wider range of conformal towers may also become available through modifying the "boundary conditions" on the entanglement cut using, e.g., rainbow chains of odd length [51].

Finally, we note that entanglement spectra may be used to study any CFT wave function, not just those obtained from diagonalising a critical Hamiltonian. This is particularly attractive for studying the edges of topologically ordered phases (e.g., chiral spin liquids): The latter are often described in terms of Ansatz wave functions obtained from, e.g., parton constructions, without an explicit parent Hamiltonian. Ansatz wave functions for the corresponding edge theories may be obtained from similar parton constructions [52,53]: Adapting these constructions to the rainbow chain would allow us to compute edge spectra in unprecedented detail, which would be particularly useful to understand novel topologically ordered phases as well as transitions between them.

**Code availability.** A TeNPy-based [45] implementation of the NRG and unzipping algorithms described in Sec. 4 is available at https://github.com/attila-i-szabo/rainbow-chain-NRG/tree/v1.0. The code used for the free-fermion calculations in Sec. 3 will be published with Ref. [34].

## Acknowledgments

I thank Natalia Chepiga, Juraj Hasik, Titus Neupert, and Hong-Hao Tu for helpful discussions. I am especially grateful to Andreas Läuchli for pointing out the similarity between rainbow chains and Wilson chains. MPS calculations were performed using the TeNPy library [45]. Free-fermion calculations also used the TeMFpy [34] library.

**Funding information** A. Sz. was supported by Ambizione grant No. 215979 by the Swiss National Science Foundation.

## A Entanglement entropy in rainbow chains

The conformal transformation (2) maps the reduced density matrix of a uniform chain of length $L$ onto the thermal density matrix of the CFT on an interval of length $w_0 \sim \log L$

[Fig. 1(a)]; as a result, the von Neumann entropy of the former equals the thermal entropy of the latter. The latter is extensive in $w_0$; the factor of proportionality is given by the Cardy–Calabrese formula [18] as

$$S = \frac{c}{6} \log L + \tilde{c} = \frac{c}{6} w_0 + \tilde{c}' . \tag{A.1}$$

**Rainbow chains.** As explained in Sec. 2, critical chains with non-uniform couplings can be viewed as a discretisation of the corresponding continuum CFT with inhomogeneously distributed reference points, separated by distances that scale as $\ell_i \sim 1/J_i$ [cf. (5)]. Therefore, upon the conformal transformation (2), their reduced density matrices map onto the thermal density matrix of a system of length

$$w_1 = -\log \tan\left( \frac{\pi}{4} \frac{1/J_{\text{middle}}}{\sum_i 1/J_i} \right) . \tag{A.2}$$

For the rainbow chain, the sum in the denominator is a geometric series, summing to

$$w_{1,\text{rainbow}} = \log \tan\left( \frac{\pi}{4} \frac{1}{1 + 2\sum_{i=1/2}^{(L-1)/2} e^{i\Delta}} \right) \simeq \frac{(L-1)\Delta}{2} - \log\left[ \frac{\pi}{8}(e^{\Delta} - 1) \right] ; \tag{A.3}$$

for $L\Delta \gg 1$, whence the entanglement entropy is given by (A.1) as

$$S \simeq \frac{c}{12} \left[ L\Delta + \ell(\Delta) \right] , \tag{A.4}$$

$$\ell(\Delta) = -2\log(e^{\Delta} - 1) - \Delta + \text{const.} \simeq -2\log\Delta + \text{const.} \tag{A.5}$$

for $\Delta \ll 1$, in accordance with Ref. [20]. This dependence on $\Delta$ comes entirely from the centre of the chain: The effective length of the bond across the entanglement cut is scaled by a factor of $\simeq \Delta/2$ to match the order of magnitude of its neighbours;[6] therefore, sites after the conformal mapping are not distributed uniformly near the corresponding end of the chain.

**Conformal chains.** For the conformal chain (7), we can no longer find $\ell(\Delta)$ in closed form. However, for $L\Delta \gg 1$, the effective segment lengths still grow exponentially except for the very ends of the chain: As a result, the effective system size only changes by a constant factor, so (A.4, A.5) will still hold, albeit with a different constant offset.

**Rings.** Similar to (A.2), the effective system size onto which the conformal ring geometry in Fig. 1(d) is mapped by the transformation (10) is given by

$$w_2 = -2\log \tan\left( \pi \frac{1/J_{\text{middle}}}{\sum_i 1/J_i} \right) . \tag{A.6}$$

The sum has no closed form for the conformally deformed coupling terms (12). Similar to the previous case, however, these can be replaced with exponentially decaying ones at the cost of adding a constant offset to $w_2$. The resulting geometric series again yields the volume-law scaling (A.4) with

$$\ell(\Delta) \simeq -4\log\Delta + \text{const.} ; \tag{A.7}$$

$\ell(\Delta)$ is "doubled" since now both ends of the conformally transformed chain are deformed from uniformity.

---

[6]As an extreme example, at $\Delta = 0$, all $w_i = 0$, so every site should be mapped onto $z_i = \pm 1$, leading to a zero term in the Hamiltonian across the entanglement cut. Instead, the rainbow chain reduces to a uniform one.

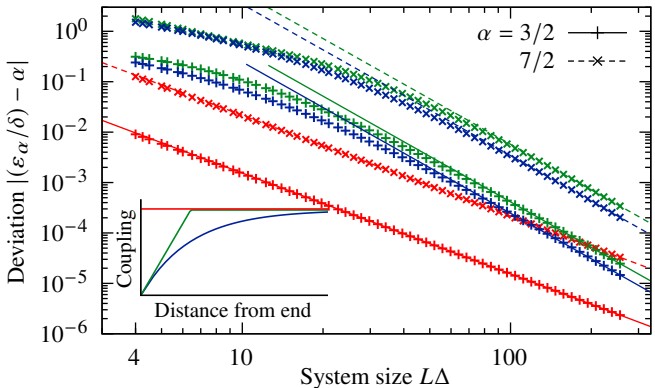

Figure 10: Log-log plot of the finite-size correction to the single-particle spectrum of a tight-binding uniform chain (red), the image of a rainbow chain ($\Delta = 1/8$) upon the conformal map (2) (blue), and a piecewise linear approximation of the latter (green). For the uniform chain, $\varepsilon_\alpha/\delta$ approaches the thermodynamic limit from below and the correction scales as $L^{-2}$; the tapered ones approach from above, scaling as $L^{-3}$ (straight lines). *Inset:* Profiles of the hopping term near the end of the three types of chain.

**Hamiltonians with single-site terms.** Both single- and two-site terms in the Hamiltonian represent the CFT Hamiltonian integrated along a segment. This effectively doubles the number of discretisation points and thus effectively halves $\Delta$ in the construction of conformal/rainbow chains. Therefore, we expect that (A.4) applies with $\ell(\Delta/2)$ replacing $\ell(\Delta)$, as indeed found numerically for the tight-binding and TFI models, see Fig. 4(d).

Finally, it is worth noting that $\ell(\Delta)$ only appears to depend on the details of the *geometry*, not of the specific Hamiltonian: Values of $\ell(\Delta)$ obtained for the TFI Hamiltonian match the entanglement entropy of the three-state Potts model accurately, as shown in Fig. 9.

## A.1 Entanglement spectrum

In addition to overall chain length, the rescaling of the central bond also affects the layout of post-conformal-transformation lattice sites. For a rainbow chain with $\Delta \ll 1$ in particular, the middle of the chain becomes nearly uniform in $z$-space, so the sites are mapped to

$$w_n = \log\tan z_n \simeq \log(n \times \text{const.}) = \log n + \text{const.}, \qquad n = \frac{1}{2}, \frac{3}{2}, \dots, \qquad \text{(A.8)}$$

resulting in segment lengths that scale as

$$\ell_n \approx \frac{dw_n}{dn} \simeq \frac{1}{n}, \qquad n = 1, 2, \dots \qquad \text{(A.9)}$$

As explained in Sec. 2, such non-uniformly discretised chains can be represented through Hamiltonian terms that scale as $f(n) = 1/\ell_n \simeq n$. That is, the entanglement spectrum of the rainbow chain is expected to match the energy spectrum of a chain with couplings that are uniform in the bulk but taper off linearly on one end.

In a very long, uniform tight-binding chain near half-filling, we expect equally spaced single-particle energy levels, at energies $\alpha\delta$, where $\delta \sim 1/L$ is the single-particle gap, and $\alpha = \pm\frac{1}{2}, \pm\frac{3}{2}, \dots$ for an even-length chain, cf. (22). Indeed, $\varepsilon/\varepsilon_{\min}$ converges to odd integers. However, while the convergence for a uniform chain is quadratic, we find it to be *cubic* for the conformally mapped rainbow chain (Fig. 10). This is entirely due to the linear tapering of hopping terms at the end, as shown by data for an additional, piecewise linear chain.

On the flipside, for $\Delta = 2$, the rainbow chain maps to a perfectly uniform chain, thus we expect that its single-particle entanglement energies (21) scale the same way as the single-particle energy levels (22) in a uniform chain. Namely, we expect that $\varepsilon_\alpha / \delta_{\text{ent}}$ approach $\alpha$ from below and that the correction scale quadratically. In numerical tests, we verified the former prediction, but we could not access long enough chains to test the scaling.

## B  Entanglement entropy and gap in free-fermion chains

As argued in Sec. 3.3, the half-chain/ring entanglement spectrum matches the energy spectrum of the image of the chain/ring under the conformal maps (2)/(10), while the entanglement entropy equals the thermal entropy of the image at inverse temperature $\beta = 2\pi$. For the chains and rings considered here, these images are uniformly spaced chains (except for the ends, which, however, do not affect the long-wavelength physics – see Appendix A.1). For $L\Delta \gg 1$, the cosine dispersions of such chains can be linearised near zero energy, resulting in fermionic modes with equally spaced single-particle energies $\varepsilon$. We define the entanglement gap $\delta_{\text{ent}}$ as their spacing, since it corresponds to a unit step in the conformal tower both for tight-binding and Majorana chains. Each of these modes contributes

$$\Delta S = -\left[ \frac{1}{1 + e^{\beta\varepsilon}} \log\left( \frac{1}{1 + e^{\beta\varepsilon}} \right) + \frac{1}{1 + e^{-\beta\varepsilon}} \log\left( \frac{1}{1 + e^{-\beta\varepsilon}} \right) \right] = \log(1 + e^{-\beta\varepsilon}) + \frac{\beta\varepsilon}{1 + e^{\beta\varepsilon}} \quad \text{(B.1)}$$

to the total entropy. In the following, we drop $\beta$, as it is absorbed into our definition of the entanglement spectrum.

**Tight-binding chain.** In the limit $L\Delta \gg 1$, i.e., that of small entanglement gap, we can replace the sum of (B.1) over all modes with the integral

$$S \simeq \int_{-\infty}^{\infty} \frac{d\varepsilon}{\delta_{\text{ent}}} \left[ \log(1 + e^{-\varepsilon}) + \frac{\varepsilon}{1 + e^{\varepsilon}} \right] = \frac{\pi^2}{3\delta_{\text{ent}}} \,. \quad \text{(B.2)}$$

**Majorana chain.** The situation is almost identical, except that modes with eigenvalues $\pm\varepsilon$ are not independent but rather creation and annihilation operators of the same Boguliubov quasiparticle. Therefore, the total entanglement entropy is

$$S \simeq \int_{0}^{\infty} \frac{d\varepsilon}{\delta_{\text{ent}}} \left[ \log(1 + e^{-\varepsilon}) + \frac{\varepsilon}{1 + e^{\varepsilon}} \right] = \frac{\pi^2}{6\delta_{\text{ent}}} \,. \quad \text{(B.3)}$$

**General Hamiltonians.** The volume-law entropy scaling (17) motivates introducing the effective chain length (18) to account for the perturbations discussed in Appendix A. In terms of this effective length, both (B.2, B.3) can be written as

$$\delta_{\text{ent}} \simeq \frac{4\pi^2}{L_{\text{eff}}} \,. \quad \text{(B.4)}$$

We anticipate [and indeed find for the three-state Potts model, see Fig. 9(b)] that this relation holds beyond free-fermion models: In a chain of length $L_{\text{eff}}$, momentum modes are discretised in units of $2\pi / L_{\text{eff}}$; due to conformal invariance between space and time, this is also the natural unit of energy. The definition of $\delta_{\text{ent}}$, however, also contains a factor of $\beta = 2\pi$, yielding (B.4).

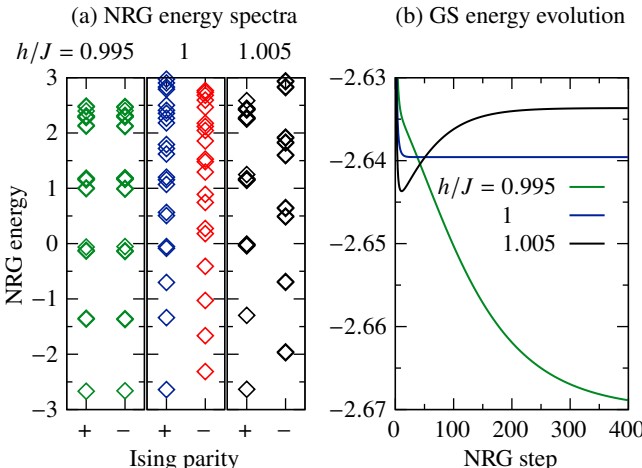

Figure 11: **(a)** Low-energy spectrum of the transverse-field Ising model (15a) after 400 NRG steps ($\Delta = 1/4$, $\chi_{\mathrm{NRG}} = 400$) for $h/J = 0.995, 1, 1.005$. The spectra are consistent with spontaneous $\mathbb{Z}_2$ symmetry breaking, the CFT spectrum in Fig. 3(a), and a paramagnetic phase, respectively. **(b)** Ground state energy as a function of NRG time. As expected, $h/J = 1$ corresponds to a fixed point; for the other two values, the spectrum drifts towards the stable fixed points linearly before saturating.

## C  NRG evolution for the transverse-field Ising model

We performed NRG simulations of the rainbow $[f(x)$ given by (8, 9b)$]$ transverse-field Ising (15a) chain at $\Delta = 1/4$, $\chi_{\mathrm{NRG}} = 400$. We found that the unrenormalised critical point, $h/J = 1$, recovers the expected BCFT spectrum accurately [Fig. 11(a)], and accordingly, the spectrum drifts negligibly little as a function of NRG time [Fig. 11(b)].

At first, we also ran NRG away from the critical point, at $h/J = 0.995, 1.005$: as expected, the NRG flow drifts to two distinct stable fixed points, corresponding to the para- and ferromagnetic phases. One would expect that the drift away from the critical point is exponential, similar to the trajectories in Fig. 8, controlled by the correlation length exponent $\nu = 1$ of the TFIM. By contrast, the deviations increase linearly in the number of NRG steps before they saturate at the stable fixed points: Since the effective length scale at NRG step $n$ is $e^{n\Delta}$, this hints at a correlation length exponential in $1/|g - g_c|$, i.e., an effective correlation length exponent $\nu_{\mathrm{eff}} = \infty$.

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
