# Peer review of "Rainbow chains and numerical renormalisation group for accurate chiral conformal spectra"

_SciPost Physics, doi:SciPost Phys. 19, 075 (2025)_

## Round 2 · Referee Report · Anonymous (Referee 1) · 2025-5-19

Strengths

1- Recently it has been shown that the introduction of inhomogeneous couplings in critical Hamiltonians can be interpreted as a spatial deformation, with the rainbow chain as a prime example of this type of analysis. This article reverses this reasoning, and claims that the analysis of deformed critical Hamiltonians can provide a route towards a detailed numerical characterization of the different conformal field theories (CFT). The idea is both simple and brilliant.

2- The article puts in practice this idea in two different setups: Gaussian states, through the use of a free-fermion Hamiltonians, and interacting systems, through a clever use of matrix product states (MPS) using the folding trick which has been put forward by researchers in the field. Also the "unzipping trick" seems to me very interesting and valuable.

3- The paper is carefully structured and written, and constitutes a valuable addition to the literature in CFT, entanglement structure in quantum many-body systems, quantum inhomogeneous systems and MPS.

Weaknesses

1- The writing style is sometimes too technical, and can not be followed easily by people who are interested in inhomogeneous quantum many-body systems with a basic knowledge of CFT.

2- The author sometimes observes unexpected behavior (please, see requested changes), but does not provide at least a hint to understand its origin.

Report

I will be very happy to recommend this manuscript for publication in SciPost Physics once the following minor modifications have been considered by the author.

Requested changes

I would like the author to consider the following minor modifications to the manuscript.

1- Sometimes the author introduces notation which might be standard in the field, but is not known to most readers. For a non-expert in the Potts model, the "Q=1 sector" is non-standard notation. Please revise the paper with this idea in mind.

2- I am intrigued by the cases in which the author does not find the expected behavior. While I appreciate the honesty and candor associated, I miss an explanation, even if incomplete. It might even go into the conclusion, as further work. Let me provide a list:

  • Page 10: why the 1/16 chiral conformal block is not found? Maybe it is a numerical problem?

  • Page 12: why do we observe the L_{eff}^{-2} scaling? It is not expected as a correction to the dispersion relation.

  • Page 17: why do we observe a \nu~16/3? As the author says, it is "vastly different" from the theoretical value, \nu=5/6.

3- I would appreciate more detail referred to the "unzipping" algorithm. I find it novel and intriguing.

4- I find the sentence "since the entanglement entropy represented by the MPS is maximised if all Schmidt vectors are equal" (page 18) extremely confusing.

5- I miss some further exploration regarding the maximal \chi_{NRG} required in order to capture the state faithfully. The author discusses low values and then fixes \chi_{NRG}=400 or 800, both cases are relevant. But, how should \chi_{NRG} scale in order to be safe, for a given L_{eff}? Page 19 mentions some discrete jumps which spoil the scaling behavior, perhaps this is the reason?

6- The NRG approach might find a precedent in the analysis of the SPT-like phase that the rainbow chain presents when the number of sites is odd, check Samos et al arxiv:1812.04869. There they provide an MPS structure in the folded regime, much in the line of the NRG approach used by the author.

Recommendation

Ask for minor revision

  • validity: top
  • significance: top
  • originality: high
  • clarity: high
  • formatting: perfect
  • grammar: perfect

Author:  Attila Szabó  on 2025-08-07  [id 5705]

(in reply to Report 1 on 2025-05-19)
Category:
answer to question

I thank the referee for their constructive comments, which are addressed below:

1- Sometimes the author introduces notation which might be standard in the field, but is not known to most readers. For a non-expert in the Potts model, the "Q=1 sector" is non-standard notation. Please revise the paper with this idea in mind.

Reply: Q=1 is indeed nonstandard notation - I meant the eigenvalue under the $Z_3$ subgroup of the full symmetry group of the Potts model. I have now clarified this on page 16.

2- I am intrigued by the cases in which the author does not find the expected behavior. While I appreciate the honesty and candor associated, I miss an explanation, even if incomplete. It might even go into the conclusion, as further work. Let me provide a list:

  • Page 10: why the 1/16 chiral conformal block is not found? Maybe it is a numerical problem?

Reply: In the ring geometry, both effective boundaries follow from entanglement cuts (see Fig. 1d), i.e., they correspond to free boundary conditions. The ground-state entanglement spectrum then consists of the 0 + 1/2 blocks. There may be tricks similar to the ones discussed for the rainbow chains to obtain the 1/16 block as well. I have not found a suitable one, and whether there is one is tangential to the main thrust of the paper, especially as the NRG does not generalise to the ring geometry.

  • Page 12: why do we observe the $L_{eff}^{-2}$ scaling? It is not expected as a correction to the dispersion relation.

Reply: The $L_{eff}^{-2}$ scaling is expected. Entanglement energies scale as $L_{eff}^{-1} + L_{eff}^{-3}$ (expansion of a sine dispersion), but the quantity of interest is their ratio, which scales as ${\rm const.} + L_{eff}^{-2}$. This is now spelled out in more detail on Page 12-13.

  • Page 17: why do we observe a $\nu\sim16/3$? As the author says, it is "vastly different" from the theoretical value, $\nu=5/6$.

Reply: This is a very important and interesting point. I have studied this in more detail, which is discussed on Page 19-20 and Appendix C. The exact value of 16/3 is not perfectly universal, but a universal value around 5-5.5 seems to underlie it. For the TFIM, I also did not find the expected scaling exponent $\nu=1$, but rather an exponentially long correlation length ($\nu=\infty$). I could not find an explanation for this behaviour, but I have highlighted it as an important item for future work.

3- I would appreciate more detail referred to the "unzipping" algorithm. I find it novel and intriguing.

Reply: I thank the referee for their interest in this algorithm. I have expanded its description substantially on Page 15.

4- I find the sentence "since the entanglement entropy represented by the MPS is maximised if all Schmidt vectors are equal" (page 18) extremely confusing.

Reply: I thank the referee for highlighting this - I have meant "Schmidt values" rather than "Schmidt vectors". I have fixed this and expanded the sentence to clarify it further.

5- I miss some further exploration regarding the maximal $\chi_{NRG}$ required in order to capture the state faithfully. The author discusses low values and then fixes $\chi_{NRG}=400$ or 800, both cases are relevant. But, how should $\chi_{NRG}$ scale in order to be safe, for a given $L_{eff}$? Page 19 mentions some discrete jumps which spoil the scaling behavior, perhaps this is the reason?

Reply: I think there is some confusion between $\chi_{NRG}$, the number of low-energy states kept in the NRG, and $\chi$, the MPS bond dimension used for the unzipping. The necessary $\chi_{NRG}$ does not depend on system size - the folded MPS tensors converge to a fixed point, which can be repeated indefinitely. The unzipping bond dimension does depend on system size, since the unzipped MPS has to represent a growing entanglement entropy. The rule of thumb is that the Cardy-Calabrese entanglement entropy must be below $\log\chi$, which limits how much entanglement is represented by the MPS. This latter is now spelled out explicitly in Eq. (27).

6- The NRG approach might find a precedent in the analysis of the SPT-like phase that the rainbow chain presents when the number of sites is odd, check Samos et al arxiv:1812.04869. There they provide an MPS structure in the folded regime, much in the line of the NRG approach used by the author.

Reply: I thank the referee for pointing out this paper. I have highlighted rainbow chains of odd length as a possible route to generating all conformal blocks of a CFT in the conclusion.

---

## Round 2 · Referee Report · Anonymous (Referee 2) · 2025-5-28

Strengths

  1. Interesting new idea on how to extract operator content from entanglement spectrum in spin chains

  2. Nice numerics supporting the analytical claims

  3. Overall very well written

Weaknesses

  1. The discussion of the different Hamiltonians considered in this work is confusing and must be improved

Report

It has long been known that, in a critical 1D system, the spectrum of the reduced density matrix is the one of the conformal field theory on a segment with some boundary conditions (as pointed out for instance by A. Läuchli in Ref. [13]). However the length of that segment is usually the logarithm of the length of the true system, and finite size effects are typically large, so that this method usually does not give access to high levels in the different conformal towers.

Here the author uses a clever trick, which is to deal with a spin chain with inhomogeneous couplings, so that the effective length of that chain is the exponential of its true length, as in the so-called 'rainbow chain'. This drastically improves the scaling of the entanglement spectrum and the identification of the different conformal towers. The author checks this idea in free fermion chains and then moves on to the interacting case, designing a numerical renormalization group method for this problem.

Overall I think the paper is very interesting and it is well written. I agree with the author that this work "opens a new pathway in an existing or a new research direction, with clear potential for multi-pronged follow-up work ".

I think the paper should ultimately be published in Scipost, however before that I request that the author improves the discussion of the specific Hamiltonians that he is using, both in the main text and in the captions of the figures. In the present form, the manuscript is hard to read because the models are not defined sufficiently explicitely. If a reader wanted to reproduce the numerical results of this paper, they would have a hard time recovering exactly which spin chain Hamiltonian was used, and this is not acceptable. See below.

Requested changes

  • Section 2 should be written more clearly. The author says "we propose discretizing the continuum CFT to sites [...]". What does this mean in practice? How does on go from a CFT to a spin chain Hamiltonian concretely?

  • The author keeps referring to Eqs. (3)-(4)-(6) as defining the equations defining the models. But how do these couplings enter the Hamiltonian exactly? Given the Hamiltonian of a critical spin chain, how should we modify it to obtain the models investigated numerically in this paper? This is rather obscure in the present version. (Maybe the author can take inspiration from papers on inhomogeneous spin chains such as [Katsura, J. Phys. A 45 115003, 2012] or [Dubail, Stephan, Calabrese, SciPost Phys. 3, 019, 2017] where this was explained more clearly in my opinion)

  • Again about Eqs. (3)-(4)-(6) : are Eqs. (3) and (6) different? Or are they the same?

  • Fig. 2 is not clear, both the figure and the caption should be improved.

  • In the discussion of the numerical procedure and of the numerical results in Sections 3.1, 3.2, and 4.1, the Hamiltonian should be written explicitly, so that the reader knows exactly what Hamiltonian is used to obtain the numerical results.

  • finally, there is a comment in the conclusion that could also be clarified: "computing entanglement spectra only requires wave functions, not necessarily Hamiltonians, at the critical point". The fact that the entanglement spectrum is a property of the wavefunction alone is obvious and has been known since the very first papers on this topic. But of course, to obtain the wavefunction numerically, usually one has to look for the ground state of a Hamiltonian. So what does the author mean exactly here?

Recommendation

Ask for minor revision

  • validity: top
  • significance: good
  • originality: high
  • clarity: good
  • formatting: excellent
  • grammar: excellent

Author:  Attila Szabó  on 2025-08-07  [id 5706]

(in reply to Report 2 on 2025-05-28)

I thank the referee for their positive assessment of the paper and their helpful comments, especially for highlighting clearer ways to present the inhomogeneous Hamiltonians used.

  • Section 2 should be written more clearly. The author says "we propose discretizing the continuum CFT to sites [...]". What does this mean in practice? How does on go from a CFT to a spin chain Hamiltonian concretely?
  • The author keeps referring to Eqs. (3)-(4)-(6) as defining the equations defining the models. But how do these couplings enter the Hamiltonian exactly? Given the Hamiltonian of a critical spin chain, how should we modify it to obtain the models investigated numerically in this paper? This is rather obscure in the present version. (Maybe the author can take inspiration from papers on inhomogeneous spin chains such as [Katsura, J. Phys. A 45 115003, 2012] or [Dubail, Stephan, Calabrese, SciPost Phys. 3, 019, 2017] where this was explained more clearly in my opinion)

Reply: I thank the referee for highlighting these issues and suggesting the above references. The Dubail et al. reference in particular has also provided a very neat argument for interpreting space-dependent coupling strengths as deformation of the space the low-energy CFT is defined on. I have largely rewritten Sec. 2 relying on these arguments. I also introduced an explicit scaling function $f(x)$ in the definition of all Hamiltonians, so it is clearer what the definition of rainbow chains etc. [now Eqs. (7-9) and (12)] refer to. While the fact that the same scaling functions are used in several different Hamiltonians is necessarily a little confusing, I believe the current presentation is significantly clearer.

  • Again about Eqs. (3)-(4)-(6) : are Eqs. (3) and (6) different? Or are they the same?

Reply: They are formally similar, but semantically different. Eq. (3) → (7) refers to a chain, so the variable n runs from 0 at the edges of the chain to (L/2-1) in the middle. Eq. (6) → (12) refers to a ring, so the equivalent variable runs from -(L/4-1/2) to +(L/4-1/2), resulting in reflection symmetry. To make this and similar distinctions clearer, these ranges are now listed in all equations where they are relevant.

  • Fig. 2 is not clear, both the figure and the caption should be improved.

Reply: Given the improved discussion of the relationship between inhomogeneous spin chains and CFTs in Sec. 2, the TFIM → Majorana chain argument is no longer needed to fix the magnitude of single-site Hamiltonian terms. Accordingly, Fig. 2 has been removed.

  • In the discussion of the numerical procedure and of the numerical results in Sections 3.1, 3.2, and 4.1, the Hamiltonian should be written explicitly, so that the reader knows exactly what Hamiltonian is used to obtain the numerical results.

Reply: The Hamiltonian was written explicitly already [Eqs. (14), (15), (24) in the current version]. As mentioned above, the inhomogeneity of Hamiltonian terms is now parameterised in terms of a function $f$, which is written explicitly for all geometries in Sec. 2.

  • finally, there is a comment in the conclusion that could also be clarified: "computing entanglement spectra only requires wave functions, not necessarily Hamiltonians, at the critical point". The fact that the entanglement spectrum is a property of the wavefunction alone is obvious and has been known since the very first papers on this topic. But of course, to obtain the wavefunction numerically, usually one has to look for the ground state of a Hamiltonian. So what does the author mean exactly here?

Reply: Wave functions consistent with conformal criticality can be obtained by other means than diagonalising a Hamiltonian. For example, Refs. [52,53] explain how to obtain ansatz wave functions for $SU(n)_k$ WZW models from parton constructions. This might be useful to study the edges of topological phases, because those are often described in terms of partons or other wave function constructions and not parent Hamiltonians. This was already explained in the rest of the final paragraph, but I made it more explicit in the new version.

---

## Round 4 · Referee Report · Anonymous (Referee 2) · 2025-8-11

Strengths

Same as in previous report

Weaknesses

No significant weakness

Report

The author carefully adressed the issues pointed out in my previous report, and he has answered my questions satisfactorily. I think the new version is clear enough now. This is nice work. I do recommend publication of these results in Scipost Physics.

Recommendation

Publish (easily meets expectations and criteria for this Journal; among top 50%)

---

## Round 4 · Referee Report · Anonymous (Referee 1) · 2025-8-13

Strengths

Same as previous report.

Weaknesses

None relevant.

Report

The author has successfully addressed my criticisms, and the paper is now ready to be published at SciPost.

Requested changes

None.

Recommendation

Publish (surpasses expectations and criteria for this Journal; among top 10%)

---

## Round 4 · Author Response

Dear Editor,

Please find attached the revised version of my paper "Rainbow chains and numerical renormalisation group for accurate chiral conformal spectra". I wish to thank the referees for their positive assessment of the paper and their constructive comments, which have greatly improved the presentation of this version. I believe that this new version is now well-suited for publication in SciPost.

I have replied in detail to all referee comments directly under the reports.

With best wishes,
Attila Szabó

---

## Round 4 · List of Changes

1. A new discussion for the relationship between inhomogeneous critical spin chains and non-uniform discretisation of CFTs, based on Ref. [25], is presented in Sec. 2, under the heading "Inhomogeneous spin chains"

  2. Inhomogeneous spin chains are presented in terms of a scaling function f(x) that can be inserted the same way into the different Hamiltonians considered in the paper.

    explicit formulas for f(x) in the conformal chain, rainbow chain, and conformal ring geometries are given as Eqs. (7), (8), (12), respectively

    the Hamiltonians in Eqs. (14), (15), and (24) are written explicitly in terms of f(x) such that the uniform critical chain corresponds to f(x) = 1

  3. Fig. 2 has become unnecessary and has been removed.

  4. Eq. (22) is changed to explicitly contain the first subleading term of the scaling of entanglement energy and the first subleading term of $\varepsilon_\alpha/\delta_{ent}$ is written out explicitly.

  5. The description of the "MPS unzipping" algorithm in Sec. 4 is expanded to describe all steps of the algorithm in more detail.

  6. An explicit construction of the MPO for the rainbow-chain Potts model is added in Sec. 4.1 (heading "Representing the rainbow chain as a matrix-product operator") and Fig. 6.

  7. The discussion of the excited-state entanglement spectrum in Sec. 4.1 is clarified in terms of boundary CFT operators.

  8. The discussion of the effective length scaling of the NRG flow (Sec. 4.2, heading "Finite \chi_{NRG} effects on NRG stability") is improved using more of the data in Fig. 8, and expanded with a study of the transverse-field Ising model in Appendix C.

  9. An explicit guide to setting the unzipping bond dimension \chi is added with Eq. (27).

  10. The outlook part of Sec. 5 is expanded:

    the relationship between the entanglement spectra of excited states and other boundary conditions is added as a direction of future work

    the usefulness of the method for e.g. parton wave functions is made more explicit

---

## Editorial Decision

published